# The invasive longhorn beetle *Xylotrechus chinensis*, pest of mulberries, in Europe: Study on its local spread and efficacy of abamectin control

**Victor Sarto i Monteys** [1,2]*, **Adrià Costa Ribes**[1], **Ivan Savin**[1,3]

**1** Institut de Ciència i Tecnologia Ambientals (ICTA), Entomology, Plants and Health, Edifici Z–ICTA-ICP Campus de Bellaterra, Universitat Autònoma de Barcelona, Bellaterra, Spain, **2** Servei de Sanitat Vegetal, DARP, Generalitat de Catalunya, Barcelona, Spain, **3** Graduate School of Economics and Management, Ural Federal University, Yekaterinburg, Russian Federation

* victor.sarto@uab.cat

**Data Availability Statement:** All relevant data are within the manuscript and its Supporting information files.

## Abstract

The invasive wasp-mimicking Tiger Longicorn Beetle, *Xylotrechus chinensis*, a potentially lethal pest of mulberry trees (Moraceae: *Morus* sp.), was first reported in Europe in 2018, although its colonization and establishment were estimated to have occurred during the year 2012 or earlier. In Catalonia the infestation spread from four towns and 44.1 km² in 2018 to 12 towns and 378.1 km² in 2020; in the studied town of Barberà del Vallès, infested trees rose from 16.21% in February 2016 to 59.29% in December 2018. Human safety in public parks and avenues is a concern, as beetle infestation increases the risk of falling branches. The main objective of this study was to evaluate how the infestation progresses over time, with and without abamectin treatment, and provide insights into female egg-laying preferences. Such knowledge helps contribute to management efforts to reduce expansion of the range of beetle infestation. Our statistical analysis shows that females prefer laying eggs on larger trees, on the highest part of trunks and on the crown base (this being more preferred than the trunk), and they do so on warmer, SW orientations rather than those facing N, NW and E. Emergence holes and gallery slits predict the spreading of infestations to new trees. An abamectin treatment (trunk injection) carried out at the end of April significantly reduced the number of new infestation. However, for maximum insecticide efficiency, the best time for treating with abamectin would be from mid-July to mid-August, when newly hatched larvae begin feeding on the phloem.

## Introduction

### Description of the insect

The invasive wasp-mimicking Tiger Longicorn Beetle, *Xylotrechus chinensis* (Chevrolat) (Fig 1), is a serious pest of mulberry trees which, if not controlled, causes their death. Its body

**Funding:** Ivan Savin acknowledges financial support from the grant from the President of the Russian Federation for young doctors of science MD-3196.2019.6. The City Hall of Barberà del Vallès (Barcelona) contributed to the costs of organizing and running this Project. The Departament d'Agricultura, Ramaderia, Pesca i Alimentació of the Generalitat of Catalonia contributed 50% of the publication costs.

**Competing interests:** The authors have declared that no competing interests exist.

is relatively large (15–25 mm in length), the antennae are short and widely separated, characteristic bands adorn the elytra and the pronotum [1]. Han and Lyu [2] provided a description of the adult along with colour pictures showing it in dorsal and lateral views, whereas the larvae and eggs were described in [1]. The larvae look typically longhorn, i.e., they have a conical-shaped trunk with very well marked pseudopoda and a whitish colour.

## Native range and invasion history

This invasive species is native to the East Palaearctic region (NE China, Taiwan, Korean peninsula, and Japan), It was first reported as settled in Europe in 2018, specifically in Catalonia (NE Spain) [1], although the colonization and establishment of this beetle there were estimated to have occurred during the year 2012 or earlier. This beetle had been intercepted before, without establishing, in Germany and the U.S.A. Indeed, in 2007, specifically in Weissenhorn, a town in the Neu-Ulm district in Bavaria, two specimens (a male and a female) appeared on June 15 and 19, 2007 in a wooden packing box from China [3]. And, in June 2011, it was intercepted in the port of Philadelphia (Pennsylvania) by the *Customs and Border Protection Agency* in non-compliant wood spools that held steel wire rope shipped from China [4]. More recently, in July 2017, Schrader [5] reported an interception in Rhineland-Palatinate of a specimen in a container holding decorative elements (birch and willow) also from China.

Sarto i Monteys & Torras [1] contributed substantial new data on *X. chinensis*' mating behaviour and egg production capacity, seasonality, larval development and damage to mulberries in Catalonia. Based on this evidence, they stated this beetle would likely spread in Europe. Indeed, very shortly after, it was reported that in spring-summer 2017, around 200 mulberry trees near the harbour of the city of Heraklion (island of Crete, Greece) were found

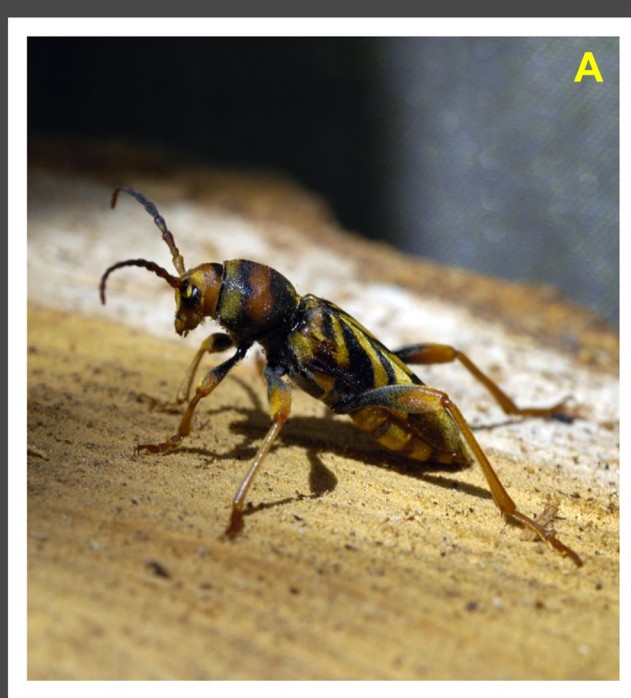
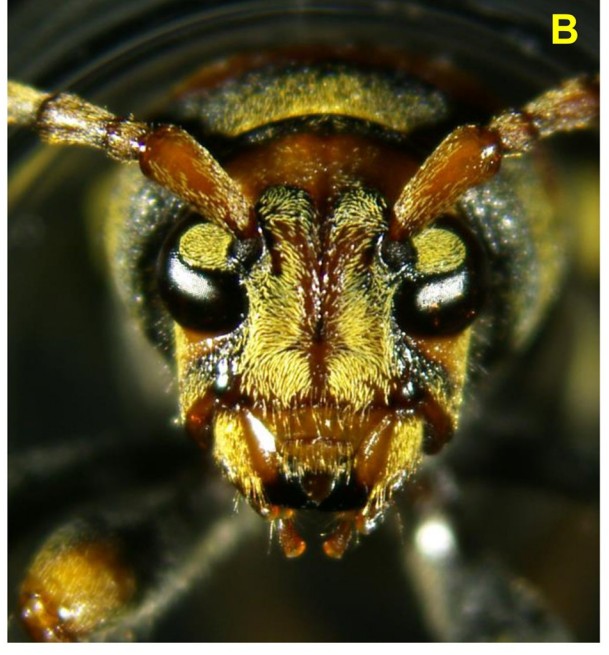

**Fig 1. A male of *Xylotrechus chinensis*.** (A) Posing alert on a mulberry bolt. (B) Close-up of male head. (Photos V. Sarto i Monteys).

to be infested by this beetle. It was most likely introduced to the Heraklion area in 2014 or 2015 with commodities arrived at this port from its area of origin [6]. In July 2018, new outbreaks were detected in the Autonomous Community of València (Spain), specifically in Quartell and Sagunt (València) and Almenara (Castelló de la Plana) [7]. Finally, in France, one cerambycid larva collected in the port village of Sète (Hérault, Mediterranean coast), in October 2017, was molecularly diagnosed as *X. chinensis*, whereas in June 2018 an adult was collected there. Likewise, in July 2018, an adult beetle was photographed at Le Bouscat (Gironde, Atlantic coast), this town being only three km away from the port of Bordeaux, which might be the entrance point for this beetle. Probably the importation of *X. chinensis* into these French towns occurred no later than spring 2017, being more likely in 2016 [8]. In Sète, the beetle was clearly established by July 2019 with at least three distinct outbreaks found and evidenced by the presence of numerous adult emergence holes seen on the trunks of mulberries [9]. French authors suppose that, given the dispersal capacities of this beetle, it has certainly already reached neighbouring municipalities and should continue to disperse. See Fig 2 for an updated map gathering all findings of *X. chinensis* in Europe up to July 2020.

In Catalonia, the beetle went from infesting one district and four townships (44.1 km$^2$) in 2018 [1] to four districts and 12 townships (378.1 km$^2$), including the city of Barcelona, in 2020 (see Fig 3). These are as follows: Vallès Occidental (Cerdanyola del Vallès, Badia del Vallès, Barberà del Vallès, Ripollet, Rubí, Sant Quirze del Vallès, Sant Cugat del Vallès, Sabadell, Terrassa), Vallès Oriental (Santa Eulàlia de Ronçana), Baix Llobregat (Molins de Rei), Barcelonès (Barcelona).

In August 2018, the European and Mediterranean Plant Protection Organization (EPPO) added *X. chinensis* to its *Alert List*, in order to draw the attention of EPPO member countries about this pest, which might present a risk to them [10]. This means that new outbreaks should be reported, and notice given to either stop or eradicate them. To date, it has not yet been declared a Quarantine pest in Europe, and therefore not subject to mandatory control measures.

## Hosts and impacts

*X. chinensis* larvae feed and bore on mulberry trees (*Morus* sp.), but apple, pear and common grape vine have also been recorded in literature as possible host plants (see reviews in [2,11]) which would require confirmation. At least for grape vine, in laboratory conditions, it was concluded that *X. chinensis* did not use it as an alternative hostplant [1].

The industry of sericulture, and its association to mulberry trees, was economically very important in Spain until the 19th century, especially in the areas of Andalusia and Murcia. However, it virtually disappeared during the first half of the 20th century in Spain and other southern European countries [12]. Mulberry trees used in the past to feed silkworms, mainly *Morus alba*, are now commonly used in many southern Mediterranean towns to provide shade and ornament to streets and avenues [13]. Their abundance in public and private places are greatly helping the spread and damage caused by *X. chinensis*. Sarto i Monteys & Torras [1] warned, referring to Catalonia, that the damage to mulberry trees as well as the economic consequences were already substantial. The beetle does not only cause the death of the trees but human safety in public parks and avenues is also a concern because beetle infestation increases the risk of falling branches and the need for rapid response by municipal authorities.

## Description of damage signs and symptoms

Sarto i Monteys & Torras [1] reported the life cycle of *Xylotrechus chinensis* in Catalonia and the damage caused to mulberry trees. At the end of spring and the beginning of summer adult

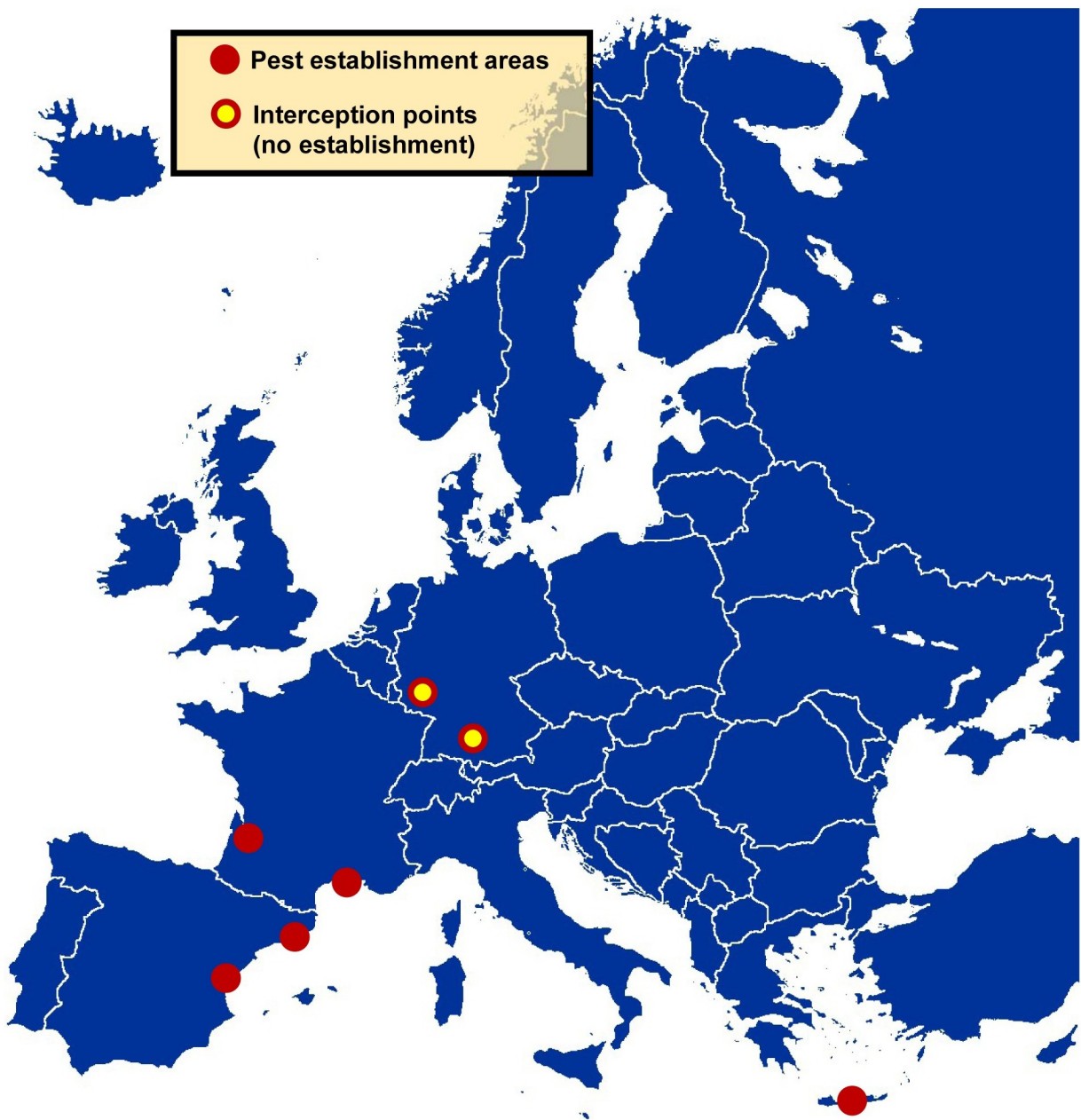

**Fig 2. *Xylotrechus chinensis* in Europe up to July 2020.** Source map reprinted from https://commons.wikimedia.org/wiki/File:Europe_map.png under a CC BY license, with permission from Wiki-vr, Public domain, via Wikimedia Commons. 30 November 2013.

beetles emerge from mulberry trees, mate, and females lay their eggs on the bark of trees. Infestation that follows can be revealed externally by two consecutive indicators, which add to the previous one constituted by the emergence holes left on the bark. Based on this evidence and new field observations, three types of indicators were selected to document infestation: (1) emergence holes, (2) bark injuries and (3) gallery slits, all visible on the trunk and crown base of trees. These indicators were counted and processed as described in the methods to inform the municipality about the progress of the beetle infestation in the study area.

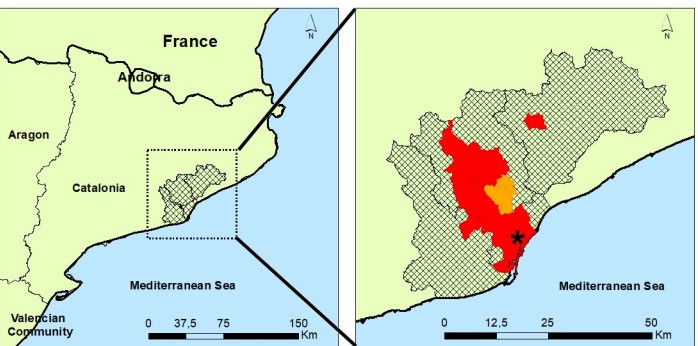

**Fig 3. Catalonian districts invaded by *Xylotrechus chinensis* up to July 2020.** The four townships invaded up to July 2018 are in orange; the eight new townships invaded up to July 2020 are in red. The black asterisk signals the city of Barcelona. Source map of Catalonia obtained from BDLJE CC-BY 4.0 Spanish 'Instituto Geográfico Nacional' http://www.ign.es/. The districts and towns were reproduced from the Cartographic and Geological Institute of Catalonia, https://www.icgc.cat/, under a CC BY 4.0 license. The data were assembled using ArcMap 10.3.1 (May 13, 2015) https://desktop.arcgis.com/es/arcmap/10.3/main/map/what-is-arcmap-.htm.

**Emergence holes (EH).**   They are round holes (*ca*. 5–6 mm diameter) on the tree periderm made by adult beetles when, after completing metamorphosis, come out from the tree from mid May to mid August. Such holes accumulate summer after summer, and remain visible as long as the periderm stays in place, even on dead trees. Fig 4A shows some of them on the crown base of an infested tree. Fig 6 below shows them along the trunk, here painted in two colours, meaning holes made in two consecutive summers.

**Bark injuries (BI).**   They are produced by active larvae, hatched annually in July-August from eggs laid on the bark, which start making feeding galleries in the tree phloem, hidden from the outside. Bark injuries only become visible when the thin periderm, that covers the cavities the larvae make, dries out and small openings or cracks appear on its surface. This only happens when the larvae have grown enough and have substantially enlarged the cavities under the periderm. Through these cracks, quite often sap oozes and some larval frass is expelled (the latter soon drying out and falling to the ground), helping to spot the injury (Fig 4B–4D). By removing the dry and dead periderm with a punch or knife (where the cracks appear), the feeding galleries can be easily discovered. These show either partly empty cavities where larvae could be found or compacted whitish yellow debris (a mixture of excrement and sawdust from the phloem consumed by the larvae). The latter may occupy, particularly in winter, the entire gallery.

Bark injuries caused by larvae hatched in summer may appear as early as October-November, depending on how quickly the larvae have developed. Occasionally, full-grown larvae have been found overwintering under bark injuries at the end of December. Bark injuries become increasingly more evident towards the end of spring when all larvae have finished growth and have bored into the xylem to pupate. Particularly in the first half of May, many bark injuries still harbour a larva in the cavity underneath (see for instance Fig 4B–4D, with photographs taken on May 10, 2016). Consequently, the presence of bark injuries can be a good indicator of infestation, although their counts underestimate the real number of larval presence in trees. Bark injuries transform into gallery slits usually one year after they appeared. Hence, unlike emergence holes and gallery slits, they are not accumulative on the trees. However, in our study, they are accumulative, since injuries seen in June have not had enough time to transform into gallery slits by December.

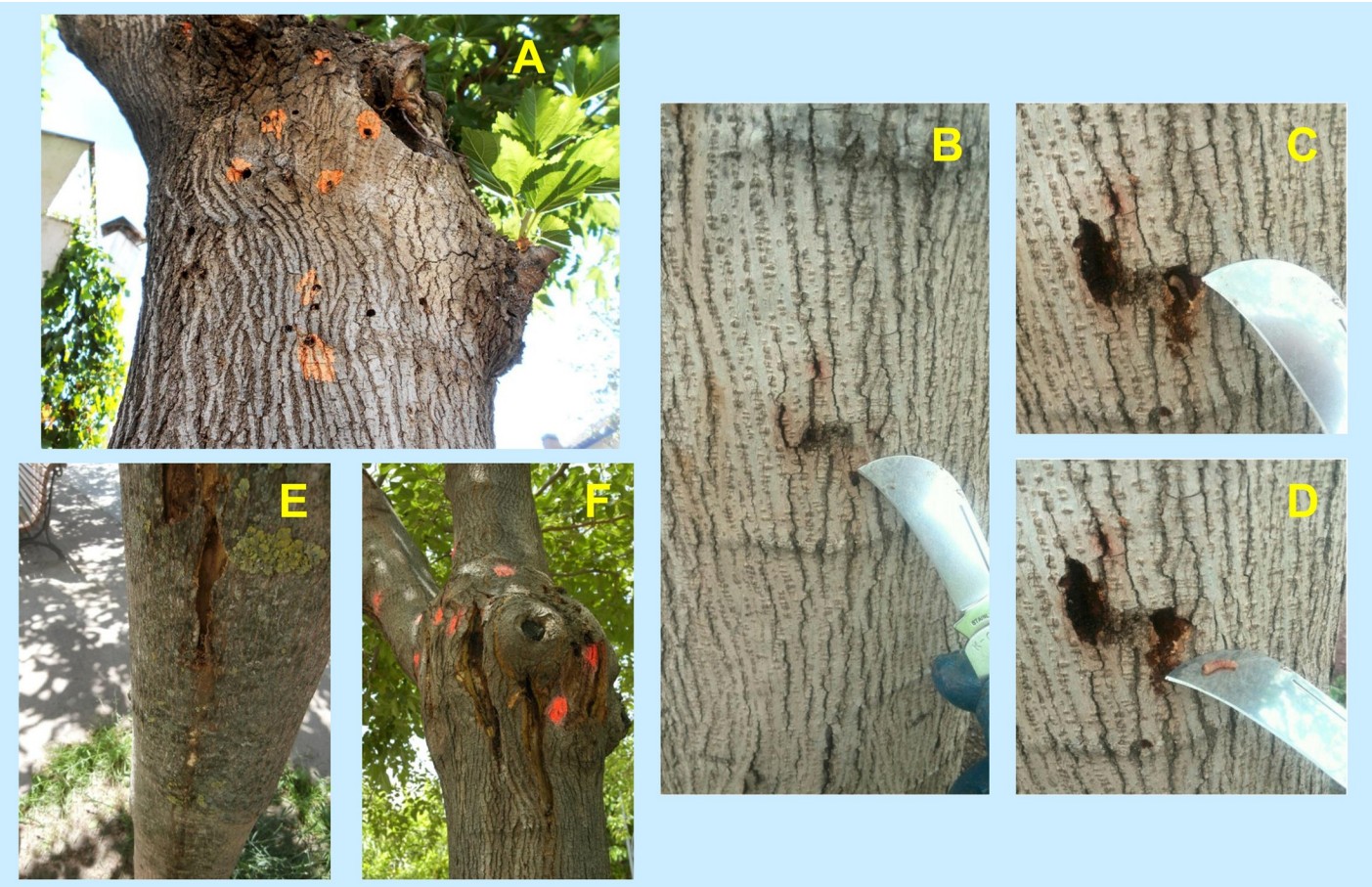

**Fig 4. *X. chinensis* infestation indicators on mulberry trees.** (A) Adult beetle emergence holes (most coloured orange) on the trunk of a tree. (Photo G. Torras). (B to D) Bark injury caused by beetle larvae: (B) as seen before removing the dead and dry periderm to discover the feeding gallery underneath, and one beetle larva (C) and (D). (Photos J. Serra). (E-F) Gallery slits caused by beetle larvae: (E) on the trunk, (F) on the crown base. Coloured spots signal emergence holes. (Photos G. Torras (E); V. Sarto i Monteys (F)).

**Gallery slits (GS).** They are formed when the periderm that covers old galleries, i.e. presently with no larvae and filled totally or partly with debris, breaks by itself longitudinally along its full length (10–40 cm). As a result, the galleries become exposed and look like big scars on the bark (Fig 4E and 4F). The debris inside, which in many cases formed a contiguous compact mass along the covered galleries, now exposed, soon dries out in contact with the air, and progressively transforms into disintegrated pieces that end up falling to the ground. If the tree survives the beetle attack, it will eventually cover and close those slits with new phloem tissue that slides, from right and left, over the exposed and hard xylem (which is bored into for pupation but not used as a food source by the beetle larvae).

## Review of within-tree distribution of larvae and pest spread

Huang [14] reported in China about the spatial distribution of overwintering (December to April) larvae of *X. chinensis* within mulberry trees. He found the larvae were distributed in an aggregation pattern, which fitted a binominal distribution model and, to a smaller extent, a Neyman model. No studies were reviewed that reported how the beetle infestation progressed

among the trees in a particular township or area between two consecutive generations, i.e., a statistical analysis of new infestation in relation to past infestation. Nor could we find studies about the degree of infestation with respect to tree size, i.e. based on trunk height and trunk perimeter, or about the geographical orientation of infestation damages and their location on tree trunk height bands and on the crown base. All this would provide important knowledge on the preferred trees for egg laying and where the eggs are most likely to be laid. A goal of the study was to determine such information to assist in scouting for infestations.

## Management options: The past and the future

Chinese [15] and Japanese [16,17] authors dealt in the past with this beetle pest. However, the control methods they proposed are obsolete and impracticable today (see [1] for review). For instance, Aruga [17], in Japan, listed three control measures for *X. chinensis*, although they are simple and applicable to most coleopteran and lepidopteran tree borers. They are as follows: (1) adults are caught and killed; (2) if frass protrusions reveal the presence of larval cavities, a suitable insecticide should be injected into these cavities; and (3) larvae can be killed by inserting metal wire inside the cavities. And Sun [15], in China, reported that the best method was blocking every (emergence) hole with mud after having injected insecticides into them, and later inserting cotton pellets impregnated with insecticides into these same holes, being the more effective the organophosphate insecticides phoxim and chlorpyrifos.

However, injecting insecticide into the emergence holes seen on mulberries, apart from being very time-consuming, would not be efficient since they are not really well connected to the phloem cavities inhabited by larvae. Obviously, an emergence hole leads to an empty tunnel where the larva pupated and the hole is only made when the adult abandons the tree [1]. And phoxim and chlorpyrifos are presently banned for use on crops in the European Union.

Based on all data gathered, Sarto i Monteys & Torras [1] suggested possible control methods to hinder the spread of this pest. In sum, they were as follows: (a) Physical control. Removal and destruction (burning or grinding) of heavily infested mulberry trees, (b) Chemical control using insecticides, (c) Pheromone control, since the three components of the male-produced aggregation pheromone of *X. chinensis* are known, and (d) Biological control using parasitoids, since at least one likely parasitoid wasp was found by these authors.

With regard to chemical control, these authors suggested two methods that might complement each other. (1) Targeting ovipositing females, their eggs and first instar larvae, by spraying authorised contact insecticides on barks at the beginning of June, July and August. However, special regulations applying to insecticide use in public areas may hinder its implementation, and (2) Targeting larvae (all instars) feeding in the tree phloem. This would require the use of a systemic insecticide, for instance abamectin, mostly a stomach poison, introduced into the xylem through endotherapy, i.e., the trunk injection of a sap-compatible solution of the insecticide. The xylem sap moves upwards through the vessels and the active ingredient is so transported to the phloem, where beetle larvae feed. This confines the applied insecticide only within the target tree, thereby making it particularly useful in urban situations. Abamectin has shown good results against bark beetles (see for instance [18]).

Since the use of pheromone and biological control methods are not ready yet (research is still necessary in this regard), while removal and destruction (burning or grinding) of heavily infested mulberry trees (the most used measure in Catalonia so far) has been proven to be insufficient and ineffective to control the spread of this pest, some townships tried chemical control injecting abamectin to some of the trees. However, not knowing exactly the best time to treat and with no follow-up of the efficacy of such treatments, no clear guidelines for action

have been obtained so far to effectively control this pest. Our results aim to some extent to remedy this gap.

In sum, there is a need for (1) improved management to control this pest, (2) much precise knowledge of spread, (3) knowledge of within-tree distribution of larvae and females' preferences to lay their eggs on particular trees or sections of trees. Specifically, to aid in scouting, it is important to know if (a) trees with bigger trunk perimeter and height were more attractive to female beetles than smaller ones, and (b) if female beetles showed more preference for particular height bands and compass orientations to lay their eggs on the trees. Finally, (4), there is a need for results concerning the efficacy of insecticide treatments. Providing answers to the four above-mentioned points were our goals in this work.

## Materials and methods

### Surveys: Location, dates and trees

The town of Barberà del Vallès, 8.3 km$^2$, located in the district of Vallès Occidental, 14 km north of Barcelona (Catalonia, Spain), held in February 2016 506 mulberry trees spread over its public avenues, streets and squares. 16.21% of these trees were already infested by *X. chinensis* [1] (Fig 5). In 2018, two surveys were conducted under permissions granted by the City Council, checking all public trees for beetle infestation. One was in the first half of June, just before the expected emergence of the new annual generation of beetles, and the other in the second half of December, when larvae hatched during the summer (from eggs laid by those beetles) were overwintering (many already fully grown) in the trees.

In May 2018, only 438 public mulberry trees remained, since 68 had been removed (either cut, pulled out or replaced by other non-mulberry trees) by local authorities as they were in poor condition due to beetle attack and there was risk of falling branches. Therefore, the survey in June was done on 438 trees. Between June and December 2018, 30 more mulberry trees disappeared (16 had been cut, 3 pulled out, and 11 replaced by either Freeman's maple, *Acer × freemanii* or *Koelreuteria panniculata*). Therefore, the December survey was done on 408 trees.

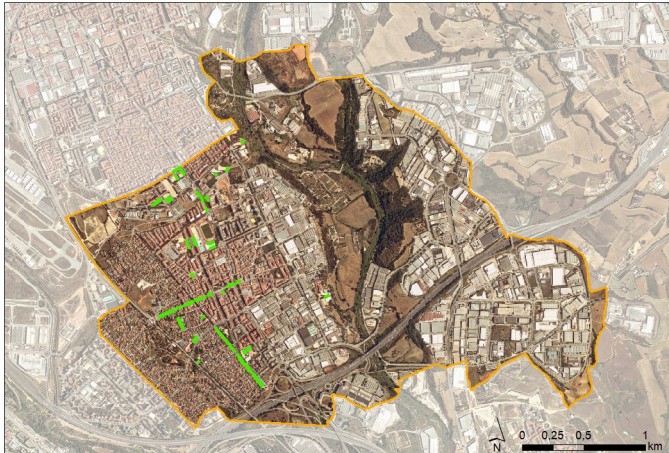

**Fig 5. Satellite map of the town of Barberà del Vallès showing in green the location of its 506 public mulberry trees in 2016.** (Map source: Cartographic and Geological Institute of Catalonia, https://www.icgc.cat/, under a CC BY 4.0 license, accessed 27 April 2020).

## Data sampling and measurement

All 438 public mulberry trees present in June 2018 were given a numerical code and located on a satellite map, so they could be recognized individually in both June and December surveys. Most trees were on a straight line, along avenues and streets, spaced about 6.5 meters. Two people working together checked each tree, acting in an orderly manner from tree 1 to tree 438, and counted their emergence holes (EH), gallery slits (GS) and bark injuries (BI), noting also their heights above the ground and their compass orientations as one of 8 directions: N, NE, E, SE, S, SW, W, or NW. Position was recorded on consecutive 50 cm-height bands along the trunk (from ground level to its top), plus a final 50 cm band comprising the lowest part of the crown, i.e. the crown base. Above these bands, infestation is generally low and it was not considered. The number of bands along the trunk varies according to its height: the taller the tree, the bigger the number. The top band on the trunk, i.e. the one that reaches the crown base, has obviously a variable height, shorter than 50 cm (Fig 6).

In addition, the trunk perimeter (at a height of 130 cm above ground level) and the trunk height from ground level to the base of the crown, i.e. when the main branches start projecting out from the trunk, were also measured per each tree using a girthing tape.

## Insecticide treatment using abamectin

On April 24 2018, an endotherapy treatment with abamectin was carried out by an external company hired by Barberà del Vallès' City Council. 107 mulberry trees were treated, nearly all lining up along one row (sometimes two) of a wide avenue heavily infested by the beetle. When two rows were present, they were spaced about 7 m. Within these rows, most trees were injected, regardless of their infestation level; their codes are indicated in S1 Table. The timing and relevance of this treatment had not been decided by us. Neither control group had been taken nor scientific follow up had been planned. This treatment was done as a response to citizen pressure to try to save the trees. Because these 107 trees, among the 438 checked in June 2018, were still present in December, an analysis of the treatment results was included in the study.

The dose of insecticide per drilled hole (trunk injection point) and drillings per tree are both determined according to the trunk perimeter. In this case, 10 ml of a mix containing 5 ml of abamectin 1,8% [EC] w/v (Vertimec®, register # 16784, Syngenta España, Madrid, Spain) and 5 ml of abamectin emulsifier (Endotree Mixable®, Endoteràpia Vegetal, Castelló d'Empúries, Girona, Spain) were injected to the tree per drilling. Typically, there were 2–3 drillings per tree. Specifically, drillings for injecting the insecticide are determined at the rate of one every 30 cm of trunk perimeter at a height of 130 cm. They are performed at the base of the tree trunk, between ground level and the first 20–25 cm, with a 7 mm drill bit, and evenly distributed around the perimetral circumference. Finally, before injecting the insecticide, a special catheter, which acts as a plug (Endoplug®, Endoteràpia Vegetal, Castelló d'Empúries, Girona, Spain) is placed in each hole. This catheter allows the product to enter through a special injection needle (external diam. 1.6 mm, internal diam. 0.8 mm, total length 80.50 mm) and prevents it from exiting when this needle is removed. It also has the function of protecting the hole against the entry of external pathogens. The equipment used for the injection is called Endoplant® (Endoteràpia Vegetal, Castelló d'Empúries, Girona, Spain). It weighs less than 4 kg and consists of three parts: a central body, composed of the mechanical piston dosing pump, a joystick pistol with LCD screen for the control of the application and two bottles containing the products to be injected (see https://endoterapiavegetal.com/ca/equips/equip-endoplant/ for application details).

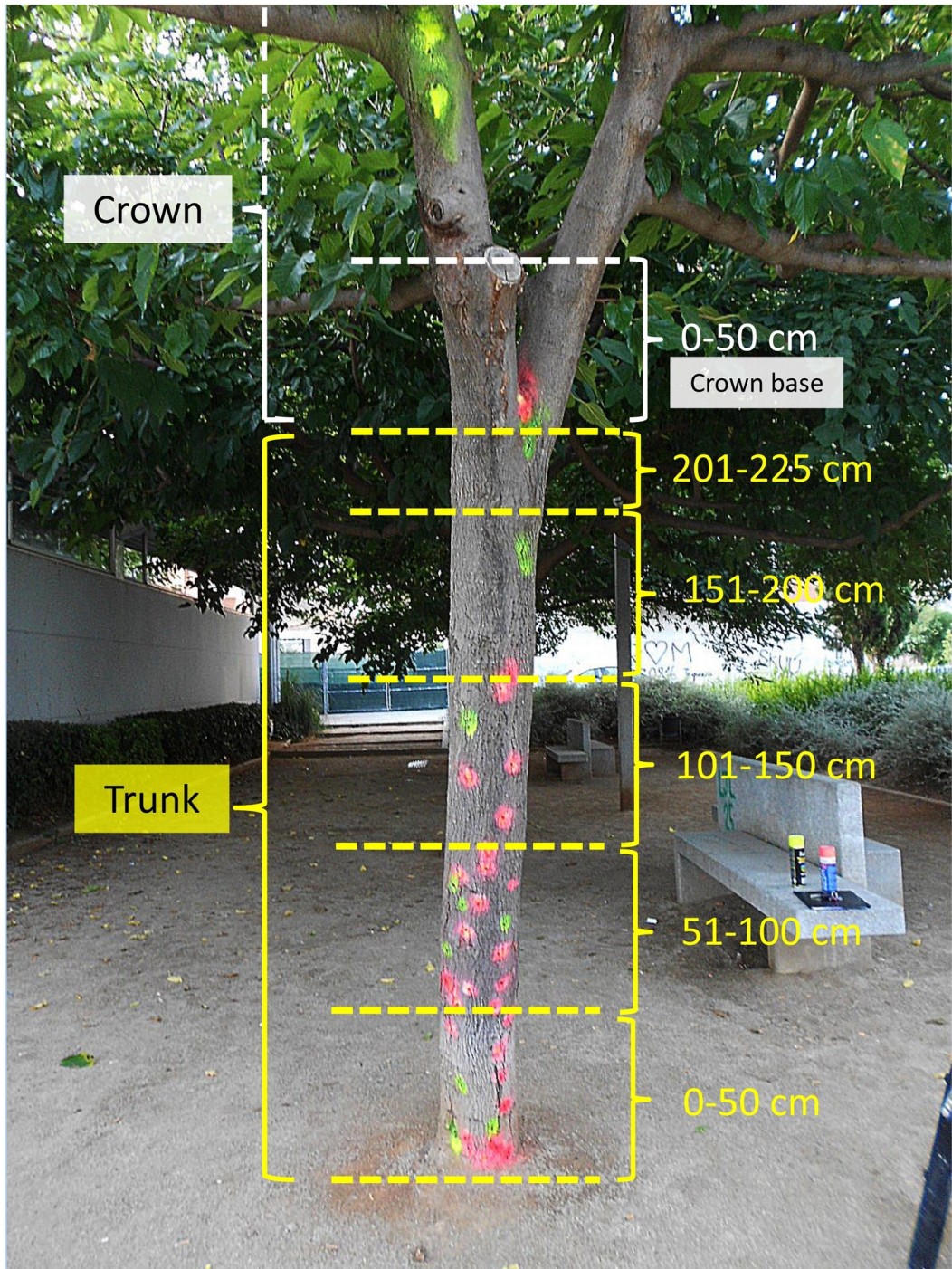

**Fig 6. A mulberry tree showing the 50 cm-height bands established for positioning indicators of infestation.** In this tree, the top band on the trunk measures only 25 cm before reaching the crown base from the fourth 50 cm-trunk band. Coloured spots signal emergence holes. (Photo G. Torras).

## Statistical analysis

All data collected for the 438 trees (408 in December) were placed in a table holding 64,824 records (S1 Table), serving as a basis for subsequent analysis. Statistical analyses of field data were performed using R software, v.3.6.2. To test for statistical differences of the distributions of infestation damages (EH, BI, GS), concerning their compass orientation and their location on tree trunk height bands and on the crown base, a non-parametric paired, 2-sample Wilcoxon Test was used that explicitly compares results for the same trees for stochastic dominance and is suitable for data distributed not normally. To test if females preferred the crown base to the trunk for laying their eggs, we used the same paired 2-sample Wilcoxon test. The null hypothesis was that the samples' mean ranks were the same (there was no difference in the number of damage for trunk and crown base). A P-value below 0.05 leads to rejection of the null hypothesis of no difference between crown base and trunk damage.

A statistical analysis of new infestations in December 2018 was conducted in relation to past infestation, height and perimeter of the tree, as well as the effect of the abamectin treatment. Because the dependent variables are count data with many zeros and conditional means have a value much below their variances, we could not use ordinary least squares (OLS) as the estimation method and opted for a negative binomial regression model (NBRM). NBRM can be considered as a generalization of the Poisson regression since it has the same mean structure and it has an extra parameter to model the over-dispersion. We also run likelihood ratio tests that suggest that NBRM is a better option (than Poisson model) for our data. Furthermore, to ensure better comparability of the control and treatment groups of trees (i.e. those treated with abamectin versus those without the treatment), we applied propensity score matching using nearest neighbour matching algorithm in MatchIt library version 3.0.2.

## Results and discussion

### Infestation of public mulberry trees from 2016 to 2018

A tree was considered to be infested when it showed at least one sign of infestation, i.e. presence of EH, BI, GS or a combination of them. The infestation of a tree can be present or past or both. In February 2016, only 16.21% of the 506 public mulberry trees of Barberà del Vallès showed signs of infestation by *X. chinensis* [1]. In June 2018, when the present study started, 68 trees had already been cut by the Municipality because of beetle attack, and of those standing (438) nearly half (46.12%) had been infested. This figure would rise to 53.36% if the 68 missing (infested) trees were included. From June to December 2018, 30 more trees were eliminated, and of those standing (408) 49.51% were infested. Again, this figure would rise to 52.97% if the 30 missing trees were included. In sum, in this town, from February 2016 to December 2018, the infestation rose from 16.21% to 59.29%, including removed trees.

The degree of infestation in trees varied notably, from only one sign up to 54 in the worst case. Table 1 and Fig 7 sort the trees checked in this study according to the signs of infestation they bore in June and in December 2018. As mentioned above, the percentage of infested trees increased in December, despite 30 trees, heavily infested in June and removed soon after, were not considered in the December counts.

Fig 8 summarizes all counts for the three established infestation indicators shown separately, comparing the June and December 2018 surveys. Since, as mentioned above, 30 trees present in June were missing in December, regression analysis will only consider the same 408 trees present in both surveys. However, we include in Fig 8 (top table) the 438 initial trees present in June to emphasize these 30 trees were contributing notably to the infestation in June, especially regarding emergence holes and gallery slits (47.05% and 25.03% more, respectively).

**Table 1. Trees sorted according their sum number of infestation indicators (EH+BI+GS) in June and in December 2018.**

| Infestation indicators' ranges | June 2018 | | December 2018 | |
|---|---|---|---|---|
| | # of trees | % | # of trees | % |
| [0] | 236 | 53,88% | 206 | 50,49% |
| [1–5] | 103 | 23,52% | 107 | 26,23% |
| [6–10] | 42 | 9,59% | 38 | 9,31% |
| [11–25] | 45 | 10,27% | 47 | 11,52% |
| [>25] | 12 | 2,74% | 10 | 2,45% |
| | **438** | **100%** | **408** | **100%** |

Should they not have been removed, the December figures would certainly have been much higher. The total accumulative figures for the three infestation indicators used in this study are 2,103. They include all counts made in June 2018, even those on the 30 trees later eliminated, plus counts of (only) new signs seen in December 2018, and distribute as follows: 943 EH (44.84%), 964 GS (45.84%) and 196 BI (9.32%).

Some discussion follows concerning these results. As to **emergence holes**, since *X. chinensis* is univoltine and there are no records of specimens enduring two-year cycles as reported for

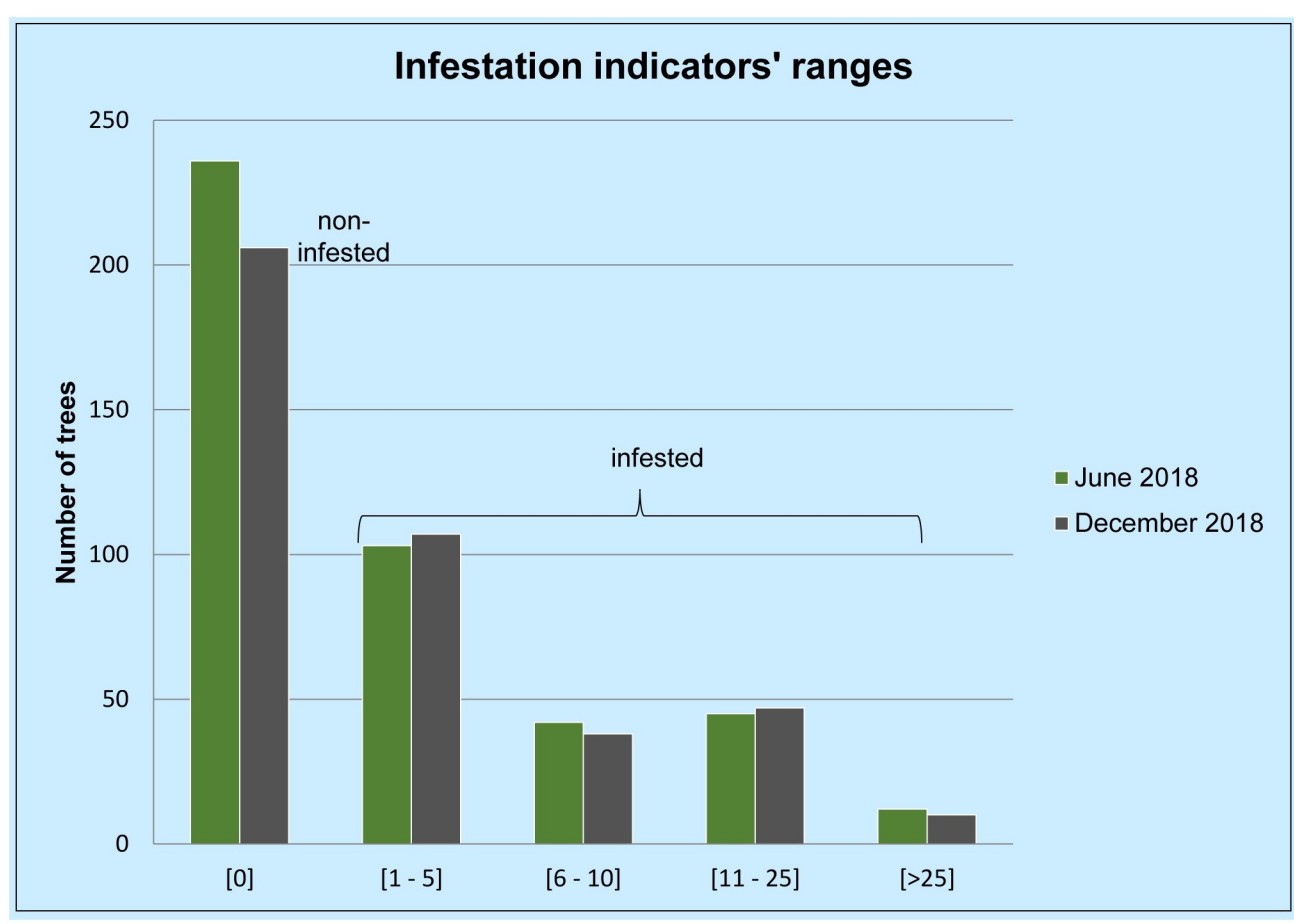

**Fig 7. Graph sorting trees according their sum number of infestation indicators (EH+BI+GS) in June and in December 2018.**

| June 2018 (438 trees) | | | | | |
|---|---|---|---|---|---|
| **EMERGENCE HOLES** | | **BARK INJURIES** | | **GALLERY SLITS** | |
| Holes on the trunk | Holes on the crown base | Bark injuries on the trunk | Bark injuries on the crown base | Gallery slits on the trunk | Gallery slits on the crown base |
| 478 (68.58%) | 219 (31.42%) | 59 (44.03%) | 75 (55.97%) | 585 (63.04%) | 343 (36.96%) |
| **697** | | **134** | | **928** | |

| June 2018 (408 trees) | | | | | |
|---|---|---|---|---|---|
| **EMERGENCE HOLES** | | **BARK INJURIES** | | **GALLERY SLITS** | |
| Holes on the trunk | Holes on the crown base | Bark injuries on the trunk | Bark injuries on the crown base | Gallery slits on the trunk | Gallery slits on the crown base |
| 320 (67.51) | 154 (32.49) | 54 (43.55%) | 70 (56.45%) | 437 (58.82%) | 306 (41.18%) |
| **474** | | **124** | | **743** | |

| ↓ | ↓ | ↓ | ↓ | ↓ | ↓ | ↓ | ↓ | ↓ |
|---|---|---|---|---|---|---|---|---|
| +47.81% | **+51.90%** | +60.39% | +66.67% | **+50.00%** | +37.14% | +4.35% | **+4.85%** | +5.56% |

| December 2018 (408 trees) | | | | | |
|---|---|---|---|---|---|
| **EMERGENCE HOLES** | | **BARK INJURIES** | | **GALLERY SLITS** | |
| Holes on the trunk | Holes on the crown base | Bark injuries on the trunk | Bark injuries on the crown base | Gallery slits on the trunk | Gallery slits on the crown base |
| 473 (65.69%) | 247 (34.31%) | 90 (48.39%) | 96 (51.61%) | 456 (58.54%) | 323 (41.46%) |
| **720** | | **186** | | **779** | |

**Fig 8. Summary counts for the three established infestation indicators, comparing the June and December 2018 surveys.** For June 2018 figures include the 438 trees initially checked, as well as, again, for only the same 408 trees also present in December 2018, this for comparative reasons (see text).

other cerambycids, it must be assumed beetles emerged one summer are offsprings of beetles emerged the previous summer. As to the counts made in June 2018, since holes stay visible on the bark of trees year after year (as long as the bark does not come off), they are in fact accumulative, summing up the holes made by beetles emerged at least in the previous summers of 2015, 2016 and 2017. This is because we estimate the beetle might have colonized the town of Barberà del Vallès in the summer of 2014, so the first emergence holes on trees of this town would have appeared in the summer of 2015.

The December 2018 counts revealed 246 new EH on the same 408 public trees checked in June. Since one new EH equals one new beetle, the latter means 246 beetles emerged in 2018 from those trees (mostly in July), i.e. an increase of accumulative values of 51,90%. If we apply the male/female ratio of 1.16, and 68.2 as the estimated average number of eggs laid per female [1], the estimated new egg load laid by females during the summer of 2018 on the trees of this town would be 7,767. In addition, there is the egg load contributed by beetles emerging from infested trees held in private gardens or compounds of this town, which was not considered. At present, the mortality rate of the different stages of this beetle (eggs, larvae, pupae and adults) and its causative agents are unknown. The finding of one likely parasitoid of beetle

larvae [1] seems anecdotal for now. So, if no effective measures of control are taken, the spread of this beetle seems inevitable.

As to **BI**, those counted in June 2018 are those caused only by larvae hatched in the summer of 2017, since, unlike EH and GS, they do not accumulate year after year on infested trees (instead they transform into GS generally one year after they appeared). This explains why their counts are much lower than those recorded for the other indicators. However, in our study, counts of BI are accumulative from June to December since the injuries seen in June have not had enough time to transform into gallery slits by December, and new BI caused by larvae hatched in the summer of 2018 are added to the count. Indeed, counts made in December 2018 revealed 62 new BI on the same 408 public trees checked in June, i.e. an increase of 50%.

As to GS, the December 2018 counts revealed 36 new slits on the same 408 public trees checked in June, i.e. an increase of accumulative values of only 4,85%. However, their accumulative numbers in June 2018 were 743, by far the highest of the three infestation indicators used. As with EH, GS stay visible on the bark of trees year after year and therefore their counts are accumulative, summing up slits appeared from at least 2015, 2016, 2017 and first half of 2018. The relatively low number of new GS accounted in December 2018 could be explained because many BI seen in June had not yet transformed into GS by December.

## Infestation with respect to tree size based on trunk height and trunk perimeter

The goal was to check if bigger trees, i.e. those with bigger trunk perimeter and height, were more attractive to female beetles (to lay eggs on) than smaller ones. Since our data accumulate damages (EH, BI, GS) attributable to a few years (see above) and we expect females to keep their egg-laying preferences year after year, we included in the analysis all damages found on the 408 trees present in December 2018 plus those found on the 30 trees present in June 2018 but missing in December. All in all, analysis to check the degree of infestation with respect to tree size was carried out on a total of 2,103 damage counts.

**Infestation with respect to trunk height.** The 438 mulberry trees checked in June 2018 had a trunk height between 155 and 335 cm (μ = 240.22), most falling into the 200–250 and 251–300 ranges, as seen in Table 2 where all trees are grouped into four ranges. With regard to numbers of tree infestation, those in the 251–300 cm range were the ones most affected (64.8% showed infestation). Regression analysis was used to test if female beetles prefer to lay eggs in trees of specific heights. Tree height was not found to be a significant factor affecting the distribution of December damage compared to June (see below). Females of this beetle are good

**Table 2.  Tree infestation by *X. chinensis* based on trunk height (from ground to crown base).**

| Trunk heights' ranges (in cm) | Values accumulated up to December 2018[1] | | | |
| --- | --- | --- | --- | --- |
| | Total trees within the range | Non-infested trees within the range | Infested trees within the range | % of infestation within the range |
| [155–199] | 18 (4.11%) | 11 | 7 | 38,89% |
| [200–250] | 296 (67.48%) | 153 | 143 | 48,31% |
| [251–300] | 122 (27.85%) | 43 | 79 | 64,75% |
| [301–335] | 2 (0.46%) | 1 | 1 | 50,00% |
| | **438** | **208** | **230** | |

[1] The 30 trees missing in December were incorporated into the counts as if they were present with the same values they had in June.

flyers, so trunk height does not seem to be important to them when it comes to lay eggs on the bark, unlike trunk perimeter which might be important (see next heading).

**Infestation with respect to trunk perimeter.**   The 438 trees had a trunk perimeter (at 130 cm above ground level) between 17.5 and 138 cm (μ = 60.0), most (64.38%) falling into the 40–80 cm range (see S1 Fig). Perimeter lengths were grouped into five ranges (Fig 9). The range that gathered most trees (190) was that of 50.1–75 cm; the least abundant range was that of >100 cm with only 21 trees. About 58% of all trees within the range 50,1–75 cm and 75,1–100 cm were infested by *X. chinensis*, whereas the infestation within the other ranges was clearly lower, especially that of the range 0–25 cm with only 32% of trees infested. Therefore, should many trees be available, females prefer laying eggs on larger trees.

This is consistent with the regression results (see below), where the tree perimeter length had a positive significant coefficient predicting new EH, BI and GS. Interestingly, if we also add a squared term, it is often found to be significant but negative, implying that starting from a certain value, a tree with a larger perimeter becomes less attractive to the beetle. Fig 10 shows the projected number of EH, BI, GS and the last two combined depending on perimeter size. The projections are done using significant estimates obtained with propensity score matching reported in shaded columns in Tables 4 and 5 below. Note that, as it was shown in [19], the perimeter with maximum infestation signs from the regression can be obtained as $^-\beta/_2\alpha$ where β is the coefficient of perimeter, and α the coefficient of its squared term. As one can see, for all infestation signs except EH (here the relationship is linear) perimeter between 60 and 80 centimeters is the one where the number of new infestations is highest.

This result does not have an obvious explanation. One could be that trees with largest perimeter stand faraway from other trees and are harder to notice by the beetle. However, by locating these trees on a satellite map, the latter would only apply in a few cases. Another explanation could be that bark injuries become increasingly more difficult to spot on the bark of trees when these are large, i.e. when they have large perimeters. The latter, though, would not apply to GS.

Comparisons were also made of all damages (a total count of 2,103 for all trees) within each established range with the expected damages within this same range (i.e. under a theoretical neutral distribution of damages, where females would show no preference for laying on any trunk perimeter). Interestingly, those trees in the trunk perimeter range of 75.1–100 cm received much more damage (35.5% more) than expected, followed by those in the 50.1–75 cm range (8.8% more). The other three ranges showed less damage than expected, especially in the 25.1–50 cm range (38.3% less). This is again consistent with the inverted U-shape relationship between perimeter size and the probability to be infested stated above.

## Orientation and location of infestation damages

We also checked how all damages (EH, BI, GS) were distributed on the trees, looking at (1) their compass orientation, and (2) their location on consecutive trunk height bands plus crown base as defined above. Results include damages seen on both trunk and crown base, although the count was done separately.

The distributions of the tree types of damage caused by beetle infestation (EH, BI, GS) were plotted in their orientation on trunk and crown base, as well as their distribution on height bands. As mentioned above, all 438 trees were in this analysis, including the December 2018 data and the data contributed by the 30 trees in June that had been removed by December. Furthermore, we aggregated the information on the orientation on trunk and crown base into a single variable of orientation. Lastly, we created two new variables: BI+GS (BI and GS

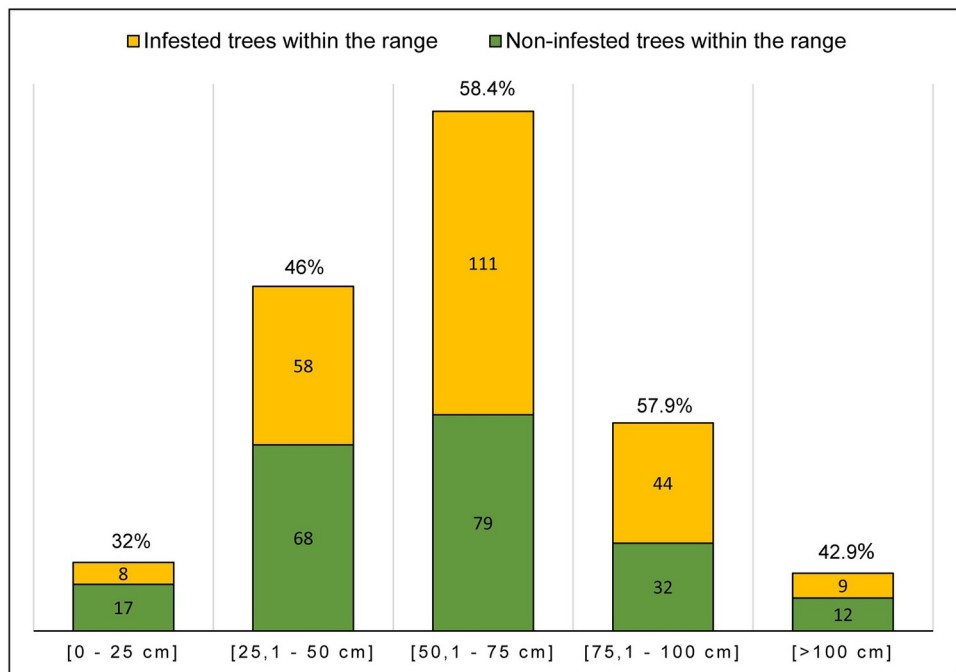

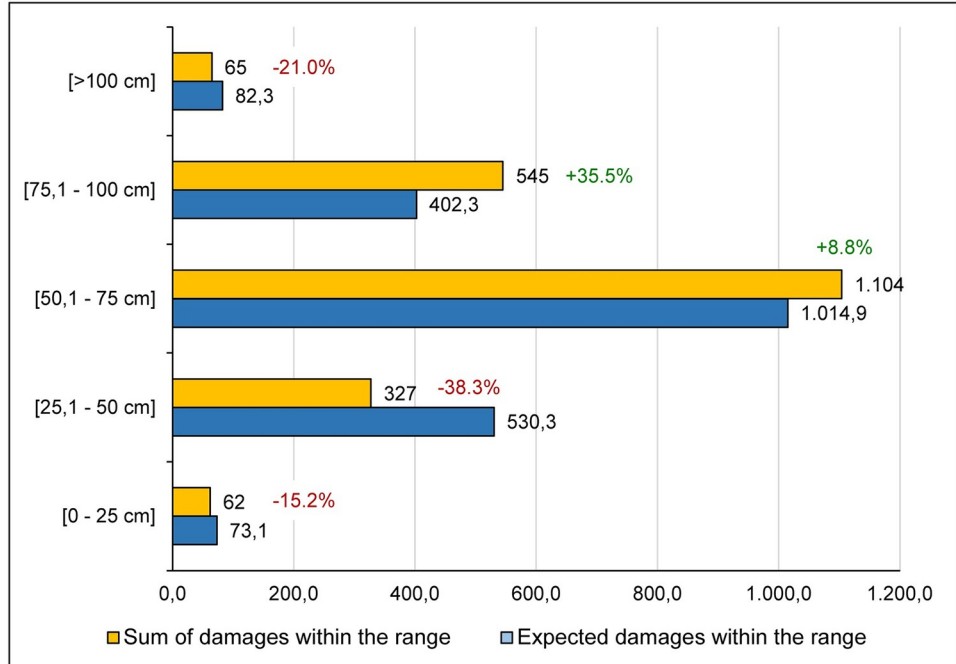

**Fig 9. Tree infestation by *X. chinensis* based on trunk perimeter.** In both graphs the 30 trees missing in December were incorporated into the counts as if they were present with the same values they had in June. The top graph shows the trees infested within each perimeter range. The bottom graph shows the sum of damages (EH+BI+GS) for all trees within their range (total damages being 2,103), and the expected damages if their distribution was neutral, i.e. with no preferences for laying on any trunk perimeter range. Percentage values in red or green tell the difference between the expected and the observed.

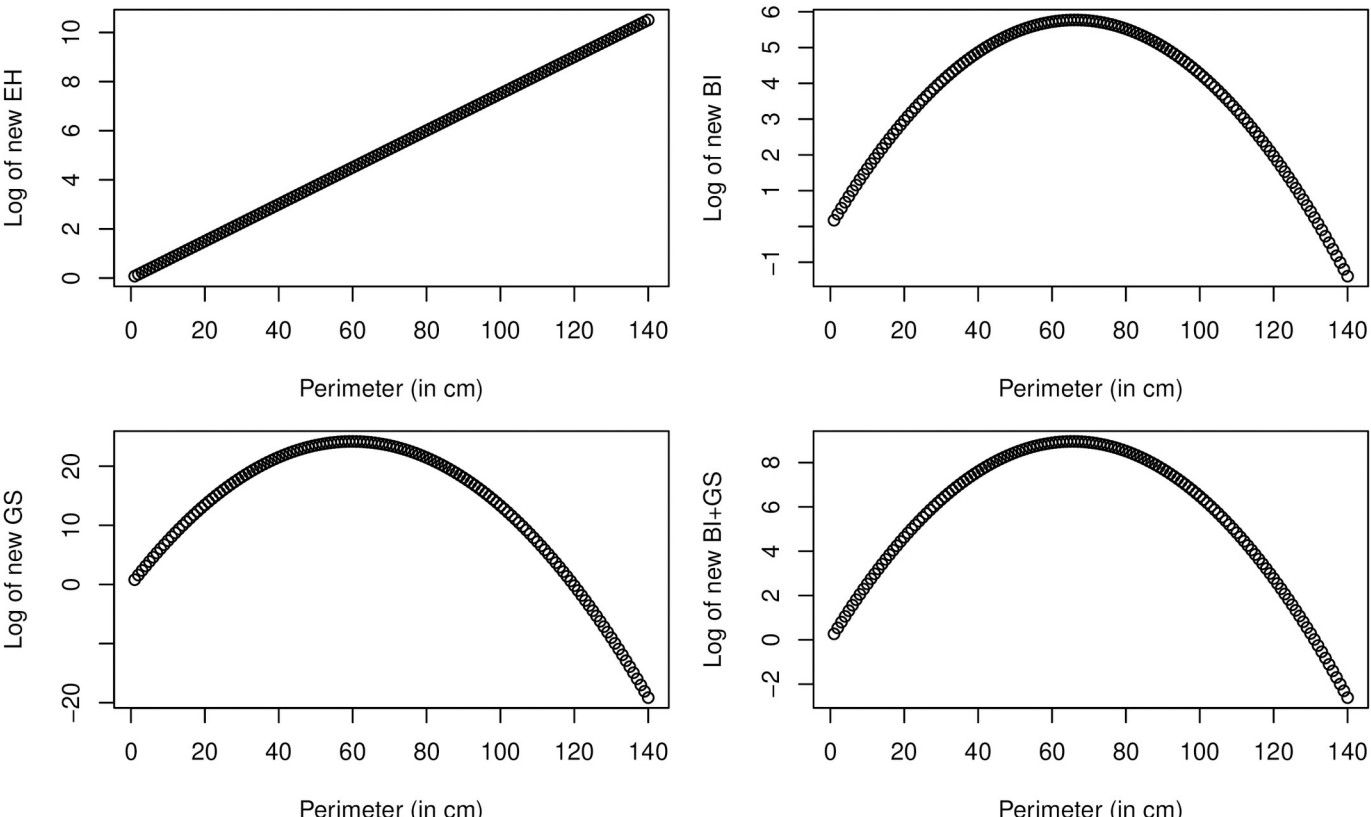

**Fig 10. Quadratic functions corresponding to the coefficients identified in the models for perimeter and squared perimeter—with other variables assumed to be zero.** BI+GS stands for bark injuries and gallery slits combined.

combined) and EH+BI+GS (EH, BI and GS combined). The goal was to form a systemic picture whether the damages are predominantly concentrated on certain orientation or height.

The results are shown in Figs 11–14, where they are summarized in two subfigures for each characteristic. The top left subfigure shows the distribution of the respective infestation sign over orientation in averages and error bars depicting +/- 2 s.e. The bottom left subfigure does the same but for the height of the infestation signs. Thus, for example, the distribution of emergence holes on NW orientation in Fig 11 has a mean of approximately 0.3 holes per tree with a confidence interval ≈0.06–0.54. The subfigures on the right side summarize results for stochastic dominance in the distribution of the given damage. All columns of the matrices represent the results of the pairwise Wilcoxon test. The dark shading indicates that the distribution of damage with an orientation or height specified in the respective row dominates the distribution in the corresponding column with a specified significance level. Asterisks ***, **, and * denote 1%, 5%, and 10% significance, respectively. Returning to the example in Fig 11, the number of emergence holes with the SW orientation is statistically higher than the one with the NW orientation, and this result is significant at the 5% significance level.

The compass distribution of EH (Fig 11) indicates that females prefer to lay eggs on the SW side of the tree, both on the trunk and crown base. The 150-200-cm and the 200-250-cm bands have higher values than the others. A somewhat similar pattern emerges in the distribution of **BI** (Fig 12). SW and NE orientations are more common, while the 0–50 cm, 200–250 cm and 250–300 cm bands are the most preferred. Concerning **GS**, N and NW side of trees are clearly

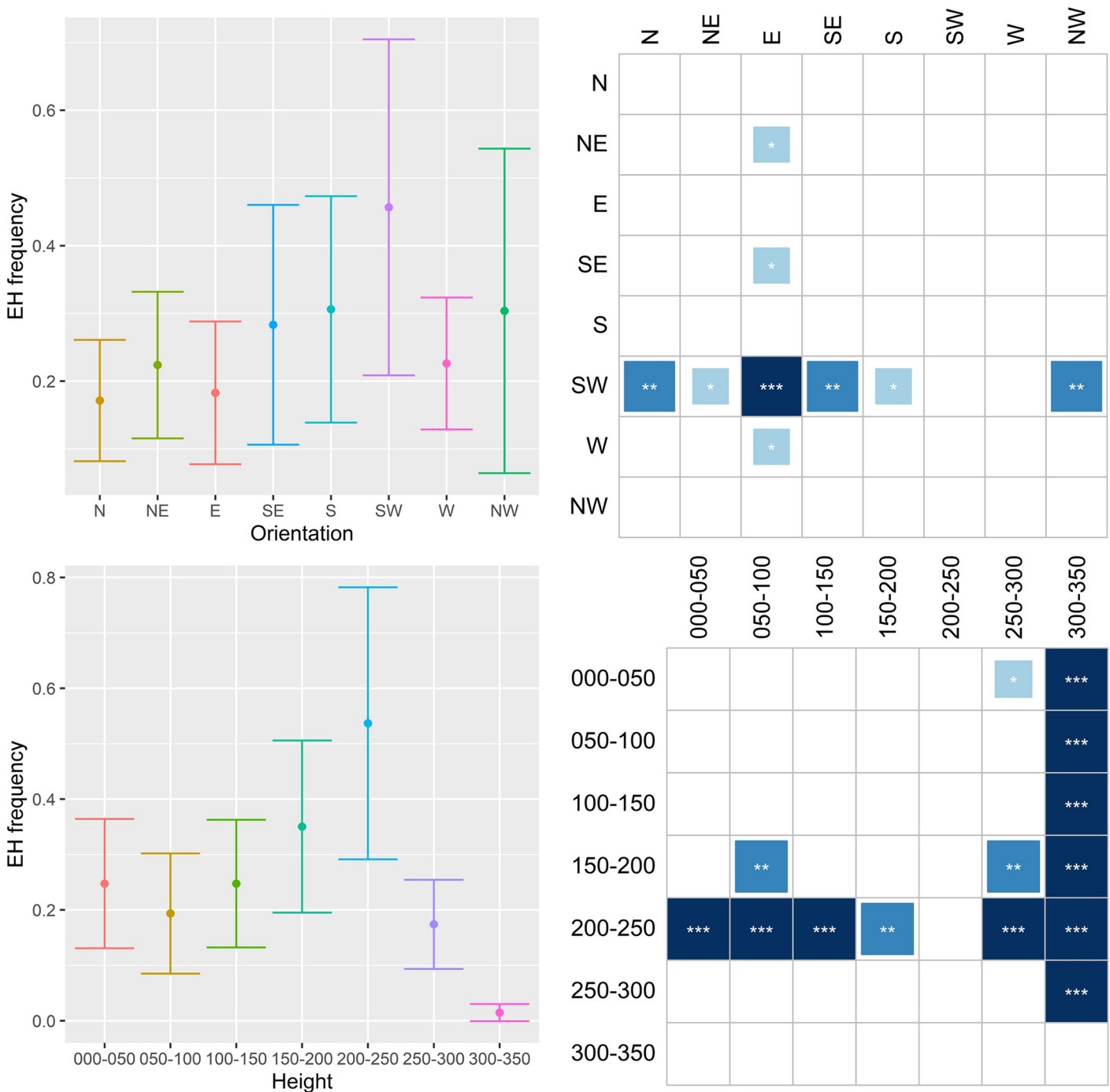

**Fig 11. Distribution of emergence holes (EH) on orientation and height.**

less preferred, while the 0–50 cm and 200–250 cm bands are the most preferred (Fig 13). When combining EH, BI and GS together (Fig 14), the SW side is the most preferred while N, E and NW the least. The 200–250 cm band shows much higher values than the others. Since larvae, after hatching, do not move much within the tree phloem, the latter means females

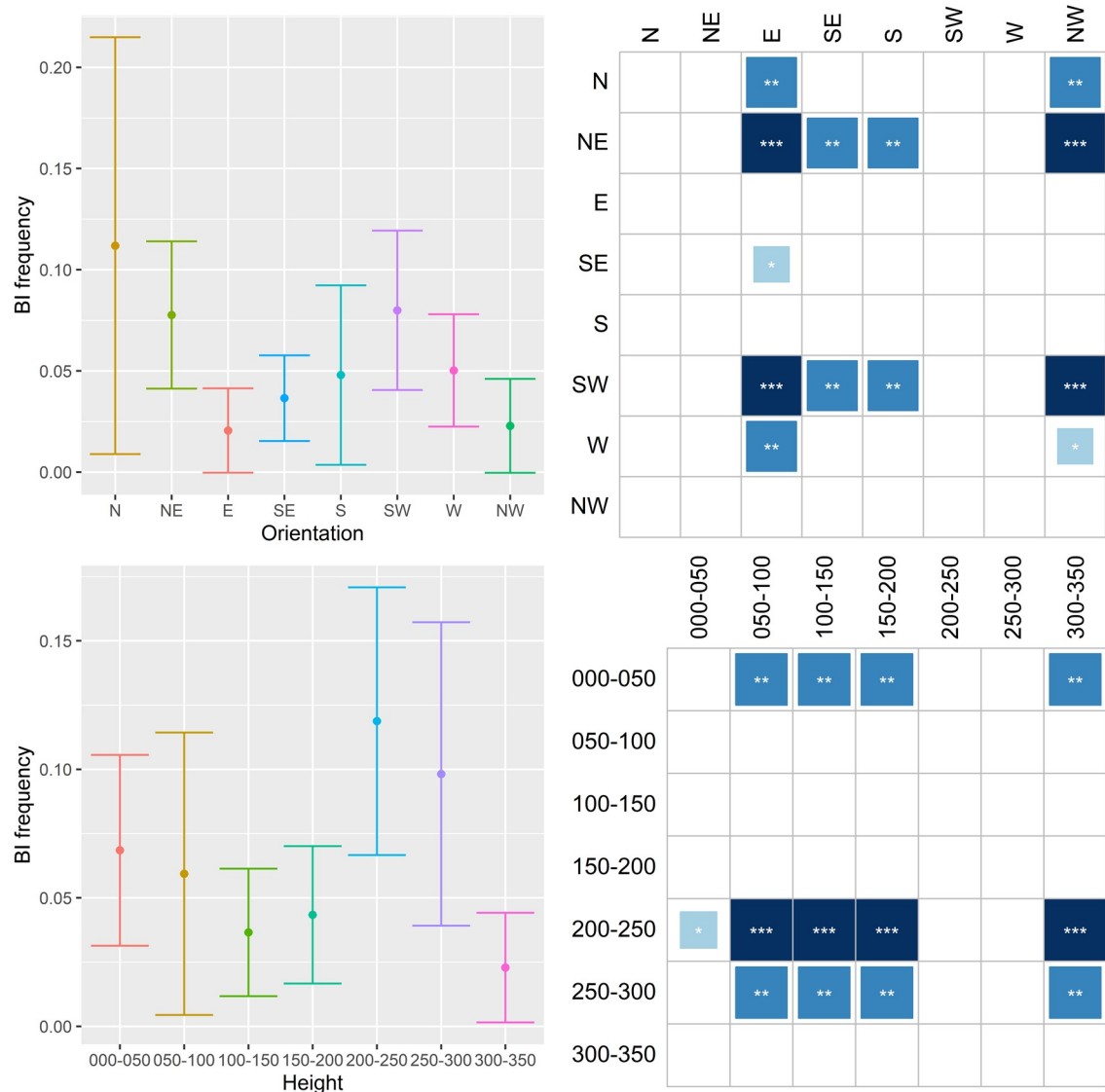

**Fig 12. Distribution of bark injuries (BI) on orientation and height.**

prefer laying eggs on the highest part of trunks and on the crown base, favoring the SW side and avoiding N, E and NW.

Finally, we checked if females preferred the crown base to the trunk for laying their eggs, since this was our perception in the field. Since trunks as a whole have a lateral surface much bigger than that of the crown base, surface values of these two pieces (trunk and crown base) should be ideally considered when checking the damages they both endured. By knowing the perimeter and height of trunks, their lateral surface can be calculated easily. The lateral surface of crown bases is more difficult to calculate, since big branches project from them, which adds surface value to the crown, yet it could be estimated as the product of trunk perimeter x 50 cm height (as defined above) x 30% more. Then, we measured the number of infestations per 1000 $cm^2$ and the results are shown in Table 3. Clearly, the number of infestation signs per surface on average is always higher for the crown base.

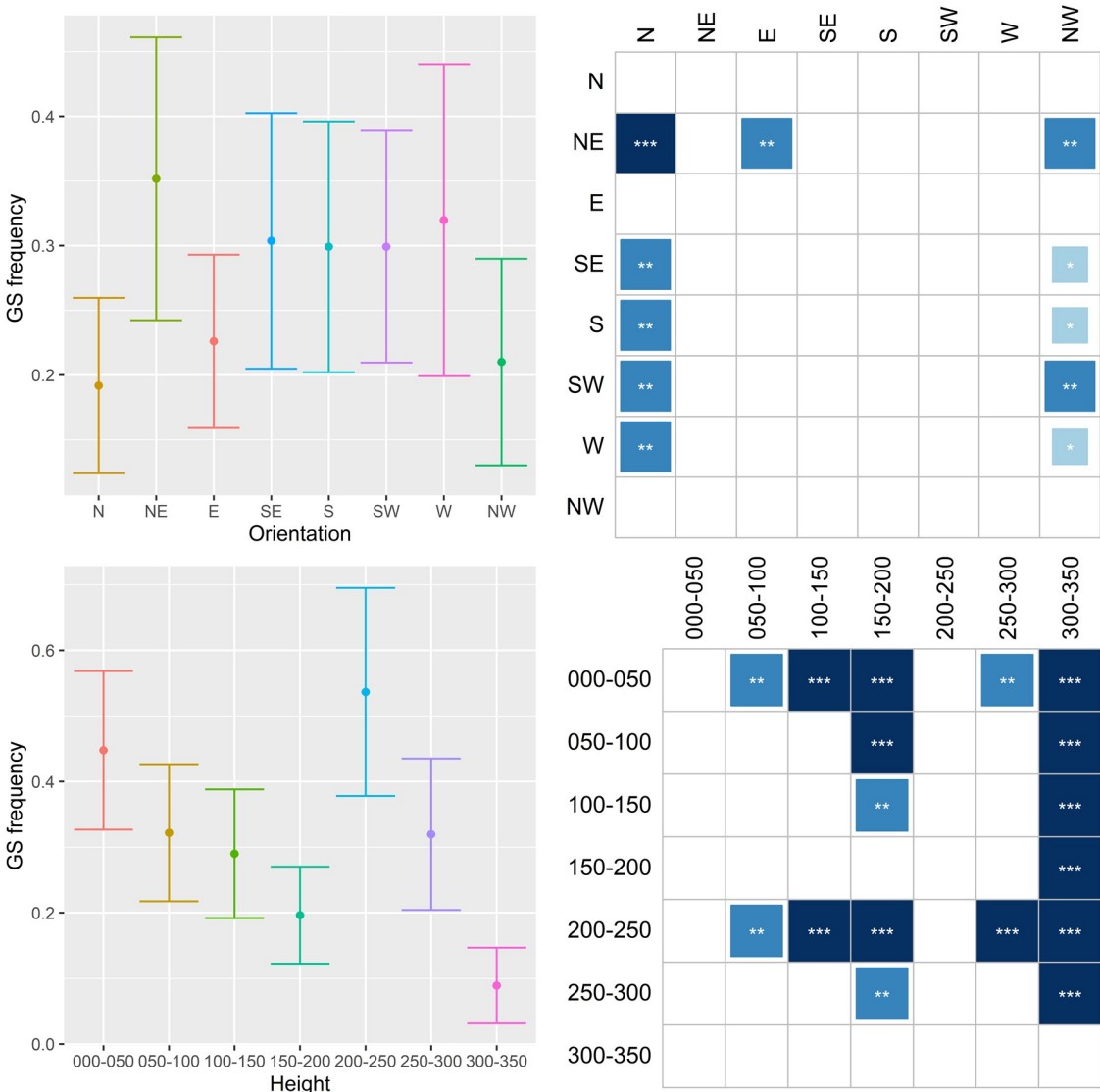

**Fig 13. Distribution of gallery slits (GS) on orientation and height.**

However, comparing the distribution of infestation signs for the same trees on the trunk and on the crown base (i.e. considering specificity for each tree) we find statistically significant results for BI in December, and for GS and all infestation signs combined (EH+BI+GS) in both months. This suggests that females prefer the crown base to the trunk for laying their eggs.

## Analysis of new infestation in December 2018 as a function of past infestation and other tree variables

By using a NBRM regression analysis, the number of new tree damages in December was explained by the damage the trees showed in June, their height and perimeter, as well as the abamectin treatment 107 trees received on April 24, 2018. In addition, the results were tested

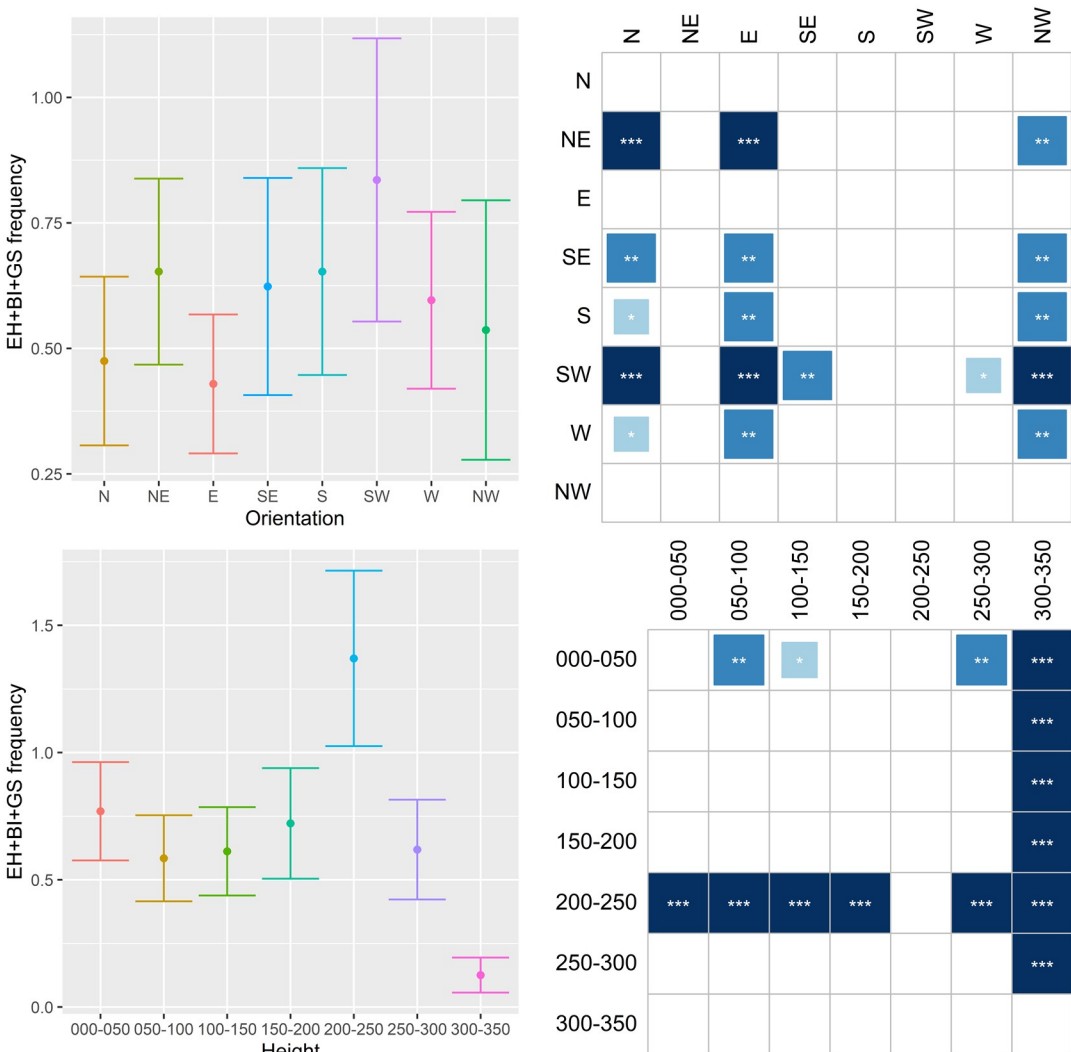

**Fig 14. Distribution of emergence holes, bark injuries and gallery slits combined (EH+BI+GS) on orientation and height.**

**Table 3. Wilcoxon test for the distribution of infestation damages on trunk and crown base (per 1000 cm$^2$).**

| Variable | Average on trunk (per 1000 cm$^2$) | Average on crown base (per 1000 cm$^2$) | Wilcoxon test p-value |
|---|---|---|---|
| EH in June | 0.0680 | 0.1279 | 0.5265 |
| EH in December | 0.0762 | 0.1464 | 0.4804 |
| BI in June | 0.0092 | 0.0521 | 0.2226 |
| BI in December | 0.0152 | 0.0683 | 0.0396 |
| GS in June | 0.1076 | 0.2081 | 0.0159 |
| GS in December | 0.0873 | 0.2117 | 0.0043 |
| EH+BI+GS in June | 0.1848 | 0.3881 | 0.0012 |
| EH+BI+GS in December | 0.1787 | 0.4263 | 0.0001 |

**Table 4. Regression results with NBRM for EH, BI and GS considered separately.**

| | ΔEH | | | | ΔBI | | | | ΔGS | | | |
|---|---|---|---|---|---|---|---|---|---|---|---|---|
| | Model 1 | Model 2 | Model 3 | Model 4 | Model 5 | Model 6 | Model 7 | Model 8 | Model 9 | Model 10 | Model 11 | Model 12 |
| Constant | 0.682 | 1.793 | -0.482 | -0.804 | -5.991*** | -10.901*** | -1.175 | -6.671 | -1.723 | -12.019** | -1.545 | -25.821** |
| EH_june | 0.223*** | 0.466*** | 0.185*** | 0.379*** | -0.355* | -0.348 | -0.401 | -0.444 | -0.112 | 0.418 | -0.504 | 9.378 |
| (EH_june)$^2$ | | -0.009*** | | -0.007** | | 0.006 | | -0.008 | | -0.045 | | -9.291 |
| BI_june | 0.074 | 0.070 | -0.038 | -0.065 | 0.212 | 0.154 | 0.073 | 0.051 | 0.214 | 0.032 | 0.182 | 0.073 |
| GS_june | 0.190*** | 0.533*** | 0.205*** | 0.306* | 0.210*** | 0.268* | 0.130* | 0.064 | 0.124 | 1.525*** | -0.031 | 1.381*** |
| (GS_june)$^2$ | | -0.028*** | | -0.009 | | -0.659 | | 0.004 | | -0.196*** | | -0.178** |
| **Abamectin** | **-0.624** | **-1.053*** | **-0.644** | **-0.854*** | **-1.316**** | **-1.346*** | **-1.354**** | **-1.298**** | **-0.070** | **-0.288** | **-0.403** | **-1.109*** |
| Perimeter | 0.026*** | 0.013 | 0.027** | 0.075* | 0.019** | 0.189*** | -0.004 | 0.174** | -0.001 | 0.421** | -0.017 | 0.808** |
| (Perimeter)$^2$ | | -0.001 | | -0.001 | | -0.001** | | -0.001** | | -0.004** | | -0.067* |
| Height | -0.016** | -0.020*** | -0.010 | -0.016* | 0.011 | 0.010 | 0.001 | -0.001 | -0.004 | -0.014 | 0.003 | -1.247 |
| AIC | 566.76 | 563.90 | 359.60 | 360.32 | 316.60 | 313.23 | 188.63 | 189.11 | 219.87 | 191.21 | 147.86 | 119.64 |
| N obs. | 408 | 408 | 198 | 198 | 408 | 408 | 198 | 198 | 408 | 408 | 198 | 198 |
| Likelihood ratio test | 290.64*** | 266.06*** | | | 30.87*** | 25.56*** | | | 32.55*** | 9.71*** | | |

Asterisks ***, **, and * indicate 1%, 5%, and 10% significance level, respectively. Results on a subsample based on propensity score matching are presented in shaded columns. A significant value of the Likelihood ratio test indicates that NBRM, estimating the dispersion parameter, is more appropriate than the Poisson model.

for the sum of BI+GS new damage. For proper comparison this analysis is limited to 408 trees present both in June and December 2018. Tables 4 and 5 below summarize the results. The coefficients in these tables should be interpreted as follows: the variable EH_june has a coefficient of 0.223 on the number of new EH in December (ΔEH, see Model 1), which is statistically significant at 1% level. This means that for each one-unit increase in EH_june, the expected log count of ΔEH increases by 0.223. In other words, each extra emergence hole in June leads to $e^{0.235} \approx 1.26$ more emergence holes in December.

**Table 5. Regression results with NBRM for BI and GS combined.**

| | Δ(BI+GS) | | | |
|---|---|---|---|---|
| | Model 13 | Model 14 | Model 15 | Model 16 |
| Constant | -3.528** | -9.495*** | -0.579 | -9.091** |
| EH_june | -0.224* | -0.117 | -0.448* | -0.436 |
| (EH_june)$^2$ | | -0.001 | | 0.006 |
| BI+GS_june | 0.170*** | 0.445*** | 0.116** | 0.378*** |
| (BI+GS_june)$^2$ | | -0.024*** | | -0.022** |
| **Abamectin** | **-0.803*** | **-1.156**** | **-0.967**** | **-1.350**** |
| Perimeter | 0.013* | 0.219*** | 0.008 | 0.273*** |
| (Perimeter)$^2$ | | -0.002*** | | -0.002*** |
| Height | 0.005 | 0.003 | -0.001 | -0.002 |
| AIC | 443.30 | 424.29 | 265.23 | 253.02 |
| N obs. | 408 | 408 | 198 | 198 |
| Likelihood ratio test | 46.12*** | 28.70*** | | |

Asterisks ***, **, and * indicate 1%, 5%, and 10% significance level, respectively. Results on a subsample based on propensity score matching are presented in shaded columns. A significant value of the Likelihood ratio test indicates that NBRM, estimating the dispersion parameter, is more appropriate than the Poisson model.

The results indicate that the number of GS in June is a significant predictor of new EH, GS and BI+GS combined in December, while EH in June is a significant predictor only for new EH in December. Thus, the more EH and GS a given tree had in June, the more damage the tree was experiencing in December. BI in June in contrast is not a significant predictor for new damages on the trees. This can be explained since, as mentioned above, bark injuries transform into gallery slits usually one year after they appeared, and this is why their accumulative counts are much lower than those recorded for the other infestation indicators (EH and GS). Therefore, EH and GS can be considered as useful predictors of new coming infestation on trees. The negative and significant coefficient for EH in June on ΔBI and Δ(BI+GS) stands out (models 5 and 13), but since the relation is only weakly significant (and mostly disappears after we apply propensity score matching), we ignore it.

As mentioned above, while perimeter tends to be positively and significantly related with new damages, the tree's height is insignificant. Moreover, perimeter sometimes has a nonlinear relationship with new damage with the squared term being negative indicating that a tree having a too small or too large perimeter has a lower chance to experience further infestation. A similar "inverse U-shape" relationship is also observed for EH and GS (models 2, 10 and 14) in June. This may be interpreted as beetles reducing infestation of a severely infested tree and possibly moving to a different one, although to confirm it new research is needed about the effects of chemical and visual stimuli. For EH the maximum of the U-shape curve is reached at approximately 25 EH in June, while for GS at 4 GS in June.

**Effects of abamectin treatment.** As mentioned above, on April 24, 2018, a total of 107 trees were injected with abamectin. Most of these trees were already notably infested by *X. chinensis* and our goal was to check if such a treatment had a significant effect on the larvae finishing their development within the trees, which would transform into adults and emerge soon (from mid-June to mid-August). Therefore, abamectin treatment was included as a predictor in models on the full sample and a robustness analysis was included, in which a propensity score matched treated trees against nearby untreated trees with respect to the five variables listed in Table 4. As a result, we obtained a subsample of 198 trees (99 being treated and 99 being not) that better fits the concept of random treatment of trees and supports credibility of our causal inference on the role of abamectin treatment. S2 Table shows how the trees treated with abamectin compare to the trees selected as a comparison group. After matching of propensity scores the means of the control group in terms of height, perimeter and the number of different infestation signs approach the means of the treated trees. QQ plots in S3 and S4 Figs further illustrate this, not just for the means, but for the entire distribution of the variables in question. It is true, however, that the attack density (particularly for gallery slits) of the treated trees is still higher than for the control group. This is because the number of infestation signs was indeed much higher among the treated trees, and creating a perfectly comparable matching group from the data was not possible. The conclusions are robust (see models 3–4, 7–8, 11–12 and 15–16). Abamectin is significantly reducing the number of new BI at 5% level and for EH and GS considered separately at 10% level, while for BI and GS combined at 1% level. The absolute reduction in the number of new EH/BI/GS for trees treated with abamectin ranges between 2.5 and 4 depending on the infestation indicator.

When the beetle larvae ingest or contact abamectin topically, the chemical interferes with neural and neuromuscular transmission that leads to insect death. Since most *X. chinensis* larvae stop feeding in May, or earlier, and abandon the phloem (where the abamectin acts) to enter the xylem to pupate, many larvae would not have encountered abamectin, and therefore they would have succeeded emerging as adults in the coming summer, when the injected abamectin will have faded within the tree, facilitating further infestation. The latter implies a much lower efficiency of this end-of-April treatment. Indeed, out of the 107 trees treated, 31

(28,97%) still showed new infestation signs in December, in particular new emergence holes. Yet, as mentioned above, the effects of abamectin were statistically significant, reducing the numbers of new infestations on the treated trees. It is likely that, for maximum insecticide efficiency, the best time for treating with abamectin (trunk injection) would be from mid-July to mid-August, when newly hatched larvae begin feeding on the phloem.

## Conclusions and future prospects

We have evaluated how the infestation progressed as a function of past infestation and provided new relevant insights into female beetle's preferences to lay their eggs on specific trees or specific areas of trees, hence mapping the larval distributions in them. This mapping was done by locating the infestation damages observed on the trees (their height and compass orientation), therefore providing key information to surveyors and managers on how to deal with this pest. Specifically, our statistical analysis shows that females prefer laying eggs on larger trees, on the highest part of trunks and on the crown base (this being more preferred than the trunk), and they do so on SW orientation avoiding those facing N, E and NW. We have also shown that emergence holes and gallery slits predict new coming infestation on trees. An abamectin treatment (trunk injection) carried out at the end of April significantly reduced the number of new infestations. However, managers should consider that, most likely, for maximum insecticide efficiency, the best time for treating with abamectin would be from mid-July to mid-August, when newly hatched larvae begin feeding on the phloem.

Finally, there are issues which have not been brought up in this work but may be important for the management of this pest. Indeed, based on responses of other invasive beetle pests, research is needed to detect volatiles released by mulberries, because if they were attractive would attract beetles over wide areas. Likewise, since potential attractants are not necessarily tree-based, attractants released by related longhorned beetles might also be of interest.

## Supporting information

**S1 Fig. Frequency distribution of 438 trees per trunk perimeter.**
(TIF)

**S2 Fig. Maps of Barberà del Vallès showing the location of the 99 trees treated with abamectin and the 99 trees used as control, both selected by propensity score matching.**
(PDF)

**S3 Fig. QQ plots illustrating the matching for the two groups of observations as seen in S2 Table, for EH_june, BI_june and GS_june.**
(PDF)

**S4 Fig. QQ plots illustrating the matching for the two groups of observations as seen in S2 Table, for height and perimeter.**
(PDF)

**S1 Table. Data table for all trees checked (438 in June, 408 in December).**
(XLSX)

**S2 Table. Comparison between treated and control groups of observations before and after applying propensity score matching.**
(PDF)

## Acknowledgments

Barberà del Vallès town hall gave us permission to carry out this study using its public mulberry trees, and its technicians Glòria Torras and Jordi Serra provided advice and some photographs. Lluís Olivet (Endoteràpia Vegetal) provided technical data on the equipment and products used to carry out the endotherapy treatment of mulberry trees.

## Author Contributions

**Conceptualization:** Victor Sarto i Monteys.

**Data curation:** Victor Sarto i Monteys.

**Formal analysis:** Victor Sarto i Monteys, Adrià Costa Ribes, Ivan Savin.

**Funding acquisition:** Victor Sarto i Monteys.

**Investigation:** Victor Sarto i Monteys, Adrià Costa Ribes.

**Methodology:** Victor Sarto i Monteys, Adrià Costa Ribes, Ivan Savin.

**Project administration:** Victor Sarto i Monteys.

**Supervision:** Victor Sarto i Monteys.

**Validation:** Victor Sarto i Monteys.

**Writing – original draft:** Victor Sarto i Monteys, Ivan Savin.

**Writing – review & editing:** Victor Sarto i Monteys, Ivan Savin.

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
