## [Decision Letter · Decision Letter 0]

20 Oct 2020

PONE-D-20-28834

The invasive longhorn beetle Xylotrechus chinensis, pest of mulberries, in Europe: study on its local spread and efficacy of abamectin control

PLOS ONE

Dear Dr. SARTO I MONTEYS,

Thank you for submitting your manuscript to PLOS ONE. After careful consideration, we feel that it has merit but does not fully meet PLOS ONE’s publication criteria as it currently stands. Therefore, we invite you to submit a revised version of the manuscript that addresses the points raised during the review process.

We look forward to receiving your revised manuscript.

Kind regards,

Richard Mankin, Ph. D.

Academic Editor

PLOS ONE

Journal Requirements:

2.Thank you for stating the following in the Acknowledgments Section of your manuscript:

[Ivan Savin acknowledges

financial support from the Russian Science Foundation [RSF grant number 19-18-00262].]

 [The authors received no specific funding for this work.]

3.We note that [Figure(s) 2, 3 and 5] in your submission contain [map/satellite] images which may be copyrighted. All PLOS content is published under the Creative Commons Attribution License (CC BY 4.0), which means that the manuscript, images, and Supporting Information files will be freely available online, and any third party is permitted to access, download, copy, distribute, and use these materials in any way, even commercially, with proper attribution. For these reasons, we cannot publish previously copyrighted maps or satellite images created using proprietary data, such as Google software (Google Maps, Street View, and Earth). For more information, see our copyright guidelines: http://journals.plos.org/plosone/s/licenses-and-copyright.

1.    You may seek permission from the original copyright holder of Figure(s) [2, 3 and 5] to publish the content specifically under the CC BY 4.0 license. 

Additional Editor Comments (if provided):

Dear authors,

please pay close attention to the comments of the reviewer, with which I concur. Several persons asked to review it previously have turned down the manuscript due to lack of easily accessible information; consequently, the authors should pay close attention to highlighting useful information that other persons addressing the management of this growing pest problem will obtain by reading this manuscript.

Reviewers' comments:

Reviewer's Responses to Questions

**Comments to the Author**

1. Is the manuscript technically sound, and do the data support the conclusions?

Reviewer #1: Yes

2. Has the statistical analysis been performed appropriately and rigorously? 

Reviewer #1: Yes

3. Have the authors made all data underlying the findings in their manuscript fully available?

Reviewer #1: Yes

4. Is the manuscript presented in an intelligible fashion and written in standard English?

Reviewer #1: Yes

5. Review Comments to the Author

Reviewer #1: This paper presents information about the history of Xylotrechus chinensis detections in Europe, and data on within tree distribution, changes in infestation level as measured by numbers of exit holes, bark injuries and gallery slits on Mulberry street trees that were or were not injected with abamectin in Barbera del Valles, Spain. Although the insecticide treatments were applied under contract by the city as a management operation and not part of a controlled, designed experiment, the results comparing changes in infestation signs compared to untreated trees may still provide valuable information about insecticide efficacy. Within tree distribution results may aid in visual surveys and regression analyses of infestation by tree height and diameter may help in predicting growth of infestations based on tree size. Overall, the paper presents some new and useful information and could be suitable for publication after major revisions.

The manuscript requires considerable editing throughout to improve the scientific English writing. There are many English grammatical errors, long verbose sentence construction, and in many cases colloquial language, such as “ends up killing them” or “settling down.” The manuscript could be greatly condensed by removing redundancies and improving organization.

For instance, description of signs and symptoms of infestation do not belong in the materials and methods but could be more concise and should be in the introduction. I suggest organizing the introduction with a description of the insect, native range, brief invasion history, hosts and impacts, description of damage signs and symptoms, and review of known management options including pesticide efficacy. Then state there is a need for improved management and knowledge of spread, within tree distribution, and efficacy of insecticide treatments. End the introduction with a clear Statement of objectives.

In the methods section there is no need to repeat description of signs and symptoms. Instead, immediately describe the field sites, how sampling was conducted, number of trees/samples, how they were selected, their spacing or experimental design/arrangement, and what variables were measured and analyzed. For the insecticide evaluation, provide greater detail about insecticide product used, application technique (name and supplier of product, type of injection needle/tool (its product name and manufacturer).

How were the 107 trees selected for insecticide injection? How far apart were trees? Since matched controls were not selected at the same time, but were later statistically chosen from the pool of remaining untreated trees of the 438 street trees, it is important to know how they compared to the control trees and their relative spacing and distribution at the site. If the most heavily infested trees were chosen for insecticide treatment their initial attack density may have been higher than the controls. Also, if controls were immediately adjacent to treated trees they may experience some spill over protection if larvae are killed in the treated trees reducing emerging beetles that may disperse and attack nearby trees.

It appears that signs of infestation (emergence holes, bark injuries, gallery slits) were counted per tree and not standardized as attack densities based on tree surface area (height and dbh). How was the sum of indicators determined (was it the total number of exit holes, bark injuries and gallery slits per tree?) Since bark injuries, transform into gallery slits, and are also associated with exit holes, summing and accumulating them may result in overestimating attack density by essentially double-counting.

There is a very large number of tables and figures and some could be dropped or combined. For instance, Table 2 repeats much of the same information as in Figure 8. They could be combined into a single table.

Figure 9 could be dropped since it simply shows 3 values in a pie chart that could be succinctly stated in the text.

Figure 10 may not be needed, since the number of trees by height category is given in the text, and the numbers by trunk perimeter are at least partially given in Table 3 and figure 11. These could be combined into 1 table or figure.

Figure 11 seems to include a table of values that are the same as what is plotted in the stacked bar charts below. Either present the results as a table or as a figure but not both. Also the quality is quite blurry (at least on my computer and printer).

Please improve the explanation for the subfigures in figures 13-16 depicting the matrices for the Mann Whitney test. I was unable to understand what they represent and the quality is also blurry.

It is standard to use tree diameter at breast height (dbh) in 5 or 10 cm dbh classes rather than perimeter.

It would be helpful to include a figure or table showing average ± SE number of exit holes, bark injuries, gallery slits at the time (or soon after, i.e., April or June 2018) of abamectin treatment for treated and control trees and again in December 2018 along with percent reduction and statistical differences.

It would also be helpful to include an overall summary/conclusion. How does the within tree distribution and abamectin treatment evaluation improve recommendations for survey and management?

6. PLOS authors have the option to publish the peer review history of their article (what does this mean?). If published, this will include your full peer review and any attached files.

Reviewer #1: No

---

## [Author Response · Author response to Decision Letter 0]

29 Nov 2020

Responses to the Editor and Reviewer

Title: "The invasive longhorn beetle Xylotrechus chinensis, pest of mulberries, in Europe: study on its local spread and efficacy of abamectin control”

Manuscript ID: PONE-D-20-28834 PLOS ONE

We are grateful to the editor and the referee for their positive assessment of our paper and the valuable comments, which helped us to improve the paper. Below we explain how we have addressed each comment.

Response to Journal Requirements:

We have re-checked the PLOS ONE style templates and believe our manuscript meets the Journal’s requirements.

2. Amended statements about the funding.

The Funding Statement should be updated as follows:

‘Ivan Savin acknowledges financial support from the grant from the President of the Russian Federation for young doctors of science MD-3196.2019.6. The City Hall of Barberà del Vallès (Barcelona) contributed to the costs of organizing and running this Project. The Departament d'Agricultura, Ramaderia, Pesca i Alimentació of the Generalitat of Catalonia contributed 50% of the publication costs.’

All funding-related text have been already removed from the new version of the manuscript.

3. About figures 2, 3 and 5 containing map/satellite images which may be copyrighted.

* The formerly submitted Figure 2 used a map of Europe whose credentials could not be obtained. Therefore, we have replaced it by another one. Now the caption is:

Fig 2. Xylotrechus chinensis in Europe up to July 2020. Source map reprinted from https://commons.wikimedia.org/wiki/File:Europe_map.png under a CC BY license, with permission from Wiki-vr, Public domain, via Wikimedia Commons. 30 November 2013

* The formerly submitted Figure 3 shows a map of Catalonia and a closeup of four invaded subareas within Catalonia. We did not use here any image from the internet, but created these maps ourselves. We used the software ArcMap from Arcgis. Now the caption is:

Fig 3. Catalonian districts invaded by Xylotrechus chinensis up to July 2020. The four townships invaded up to July 2018 are in orange; the eight new townships invaded up to July 2020 are in red. The black asterisk signals the city of Barcelona. Software: ArcMap 10.3 (10.3.1) (May 13, 2015) https://desktop.arcgis.com/es/arcmap/10.3/main/map/what-is-arcmap-.htm

* The formerly submitted Figure 5 shows a satellite map of the town of Barberà del Vallès. It was obtained from the Cartographic and Geological Institute of Catalonia. The geoinformation provided by this Institute is under a Creative Commons Attribution 4.0 International License (CC BY 4.0). Therefore, as stated in its webpage, when reusing its images, only its authorship should be cited; the Institute states that it is not necessary to follow any other procedure or sign any document. Now the caption is:

Fig 5. Satellite map of the town of Barberà del Vallès showing in green the location of its 506 public mulberry trees in 2016. (Map source: Cartographic and Geological Institute of Catalonia, https://www.icgc.cat/, under a CC BY 4.0 license, accessed 27 April 2020)

4. About all figures.

Since, as a result of the major revision we carried out, some figures were deleted, others were modified and several were renumbered, for the sake of clarity we will upload all figures again -as they are now referenced in the revised version of the manuscript. All figures were previously uploaded to the program PACE which ensured they met PLOS requirements.

Response to Reviewer: (our response is in italics).

Comment 1: This paper presents information about the history of Xylotrechus chinensis detections in Europe, and data on within tree distribution, changes in infestation level as measured by numbers of exit holes, bark injuries and gallery slits on mulberry street trees that were or were not injected with abamectin in Barbera del Valles, Spain. Although the insecticide treatments were applied under contract by the city as a management operation and not part of a controlled, designed experiment, the results comparing changes in infestation signs compared to untreated trees may still provide valuable information about insecticide efficacy. Within tree distribution results may aid in visual surveys and regression analyses of infestation by tree height and diameter may help in predicting growth of infestations based on tree size. Overall, the paper presents some new and useful information and could be suitable for publication after major revisions.

Thank you for your positive evaluation of our work.

Comment 2: The manuscript requires considerable editing throughout to improve the scientific English writing. There are many English grammatical errors, long verbose sentence construction, and in many cases colloquial language, such as “ends up killing them” or “settling down.”

We have improved the scientific English writing as suggested by the Reviewer. All colloquial language has been replaced (for instance “ends up killing them” has become “eventually causes their death” …), grammatical errors have been found and corrected, and sentence construction improved to make it more concise and less verbose. However, please consider that our study is a complex one and looks at different issues. Explaining it sometimes requires compound or complex sentence structures.

Comment 3: The manuscript could be greatly condensed by removing redundancies and improving organization. For instance, description of signs and symptoms of infestation do not belong in the materials and methods but could be more concise and should be in the introduction. I suggest organizing the introduction with a description of the insect, native range, brief invasion history, hosts and impacts, description of damage signs and symptoms, and review of known management options including pesticide efficacy. Then state there is a need for improved management and knowledge of spread, within tree distribution, and efficacy of insecticide treatments. End the introduction with a clear Statement of objectives.

Thank you for this suggestion. The Introduction has been organized as follows: (1) Description of the insect, (2) Native range and invasion history, (3) Hosts and impacts, (4) Description of damage signs and symptoms (moved from Material and Methods), (5) Review of within-tree distribution of larvae and pest spread, and (6) Management options: the past and the future. Also, the Introduction ends with a clear statement of objectives. Throughout the paper we tried to reduce unnecessary repetition of information.

Comment 4: In the methods section there is no need to repeat description of signs and symptoms. Instead, immediately describe the field sites, how sampling was conducted, number of trees/samples, how they were selected, their spacing or experimental design/arrangement, and what variables were measured and analysed. For the insecticide evaluation, provide greater detail about insecticide product used, application technique (name and supplier of product, type of injection needle/tool (its product name and manufacturer).

All Reviewer’s suggestions have been addressed. Now the section ‘Material and Methods’ has four subsections: (1) Surveys: location, dates and trees, (2) Data sampling and measurement, (3) Insecticide treatment using abamectin, and (4) Statistical analysis. Much greater details about the insecticide product used (abamectin + emulsifier), application technique (name and supplier of product, type of injection needle/tool, etc.) have been incorporated into the text.

Comment 5: How were the 107 trees selected for insecticide injection? How far apart were trees? Since matched controls were not selected at the same time, but were later statistically chosen from the pool of remaining untreated trees of the 438 street trees, it is important to know how they compared to the control trees and their relative spacing and distribution at the site. If the most heavily infested trees were chosen for insecticide treatment, their initial attack density may have been higher than the controls. Also, if controls were immediately adjacent to treated trees they may experience some spill overprotection if larvae are killed in the treated trees reducing emerging beetles that may disperse and attack nearby trees.

Nearly all 107 mulberry trees treated were lining up along one row (sometimes two rows) of a wide avenue heavily infested by the beetle, and spaced about 6.5 meters. They were injected, regardless of their infestation level; their codes are indicated in S1 Table. We have added this info to the revised manuscript. Also, as we wrote earlier in the manuscript, Table S2 as well as Figures S1 and S2 (now renamed to Figures S3 and S4) summarize the results on the propensity score matching. In particular, they show how the trees treated by abamectin compare to the trees selected as a comparison group. As we see, after propensity score matching the means of the control group in terms of height, perimeter and the number of different infestation signs become close to the means of the treated trees (Table S2). QQ plots in Figures S3 and S4 further illustrate this not for the means but the entire distribution of the variables in question. It is true, however, that the attack density (particularly for gallery slits) of the treated trees is still higher than for the control group. This is because the number of infestation signs was indeed much higher among the treated trees, and creating a perfectly comparable matching group from the data was not possible. We have now clarified this in the revised manuscript.

Here we list the codes of 99 trees treated with abamectin and selected by propensity score matching:

MA001 MA002 MA003 MA005 MA006 MA007 MA008 MA009 MA010 MA011 MA012 MA013 MA014 MA015 MA016 MA017 MA018 MA019 MA021 MA022 MA023 MA024 MA025 MA026 MA027 MA028 MA029 MA030 MA031 MA032 MA033 MA034 MA035 MA036 MA038 MA040 MA041 MA043 MA044 MA045 MA046 MA047 MA048 MA049 MA053 MA055 MA056 MA058 MA059 MA060 MA062 MA063 MA064 MA066 MA068 MA070 MA072 MA073 MA075 MA076 MA077 MA078 MA079 MA082 MA083 MA084 MA085 MA086 MA087 MA088 MA089 MA091 MA092 MA093 MA094 MA096 MA098 MA099 MA100 MA101 MA103 MA104 MA105 MA106 MA107 MA108 MA111 MA112 MA250 MA273 MA274 MA275 MA276 MA277 MA278 MA279 MA280 MA375 MA389 

And here we list the codes of 99 trees selected as a control group by propensity score matching:

MA037 MA042 MA050 MA051 MA052 MA054 MA080 MA095 MA102 MA127 MA138 MA146 MA147 MA154 MA155 MA158 MA162 MA163 MA168 MA171 MA173 MA175 MA179 MA180 MA185 MA187 MA189 MA192 MA194 MA195 MA197 MA201 MA203 MA209 MA213 MA216 MA217 MA220 MA223 MA225 MA234 MA241 MA243 MA244 MA245 MA247 MA251 MA253 MA259 MA261 MA269 MA287 MA290 MA291 MA293 MA298 MA301 MA345 MA356 MA357 MA358 MA359 MA368 MA372 MA374 MA377 MA378 MA379 MA380 MA383 MA384 MA385 MA386 MA388 MA390 MA393 MA394 MA397 MA399 MA402 MA404 MA405 MA406 MA407 MA409 MA412 MA414 MA415 MA416 MA417 MA418 MA419 MA421 MA422 MA428 MA431 MA432 MA433 MA440

Now, we have also listed these two 99-trees groups in a more complete Table S2 and added satellite maps to locate these 99-trees, treated and control (see new Supp. Inform. Figure S2).

Comment 6: It appears that signs of infestation (emergence holes, bark injuries, gallery slits) were counted per tree and not standardized as attack densities based on tree surface area (height and dbh). How was the sum of indicators determined (was it the total number of exit holes, bark injuries and gallery slits per tree?) Since bark injuries, transform into gallery slits, and are also associated with exit holes, summing and accumulating them may result in overestimating attack density by essentially double-counting.

We counted emergence holes, bark injuries and gallery slits per tree. And since we were more interested in knowing where on the tree these indicators appeared (i.e. lower trunk, or mid trunk, or upper trunk, or crown base, and how they were oriented on the compass), we used this system instead of standardizing it as attack densities based on tree surface area (height and dbh). Since we measured the height and perimeter at breast height (130 cm) for all trees, we could easily have calculated the surface areas of trees and then attribute attack densities to them. But we believe the way we did it provides more information about the within-tree distribution of larvae and their associated damage.

However, in one instance (when we wanted to check if females preferred the crown base to the trunk for laying their eggs, since this was our perception in the field) we did use surface areas. Since trunks as a whole have a lateral surface much bigger than that of the crown base, surface values of these two pieces (trunk and crown base) were considered when checking the damages they both endured. In this case we measured the number of infestations per 1000 cm2 and the results are shown in Table 5 of our original manuscript (Table 3 in the revised manuscript).

As for bark injuries, and they transforming into gallery slits, we believe there is a misunderstanding. As we wrote when describing bark injuries: “Bark injuries transform into gallery slits usually one year after they appeared. Hence, unlike emergence holes and gallery slits, they are not accumulative on the trees. However, in our study, they are accumulative, since injuries seen in June have not had enough time to transform into gallery slits by December.” So, we did not overestimate attack density by double-counting. We were very careful about that. Since in our study the three indicators are accumulative from June to December, we can easily know what was new in December by subtracting the June counts to those obtained in December. This can be clearly seen in excel Table S1.

Comment 7: There is a very large number of tables and figures and some could be dropped or combined. For instance, Table 2 repeats much of the same information as in Figure 8. They could be combined into a single table.

Done, we have dropped Table 2 and kept Figure 8, slightly modifying the text to accommodate this change.

Figure 9 could be dropped since it simply shows 3 values in a pie chart that could be succinctly stated in the text.

Done, we have dropped Figure 9, and succinctly stated these 3 values in the text.

Figure 10 may not be needed, since the number of trees by height category is given in the text, and the numbers by trunk perimeter are at least partially given in Table 3 and figure 11. These could be combined into 1 table or figure.

Done, we have dropped Figure 10 from the original manuscript. However, we have kept as Supp. Inform. the graph that showed the distribution of trunk perimeters as we believe it provides useful info to the readers (now as Fig S1).

Values in Table 3 (now Table 2 in the revised manuscript) refer exclusively to trunk height and those in Figure 11 (now Figure 9 in the revised manuscript) refer exclusively to trunk perimeter. Therefore, we believe combining them would be confusing. 

Figure 11 seems to include a table of values that are the same as what is plotted in the stacked bar charts below. Either present the results as a table or as a figure but not both. Also, the quality is quite blurry (at least on my computer and printer).

Done, we have removed the table and kept the bar charts in a new figure (now called Figure 9 in the revised manuscript). In this new version we have improved the quality of this figure.

Please improve the explanation for the subfigures in figures 13-16 depicting the matrices for the Mann Whitney test. I was unable to understand what they represent, and the quality is also blurry.

Thank you for the suggestion. We extended the explanation of the figures as follows: (keep in mind that in the revised manuscript Figs 13-16 have become Figs 11-14)

“The results are shown in Figs 11-14, where they are summarized in two subfigures for each characteristic. The first top left subfigure shows the distribution of the respective infestation sign over orientation in averages and error bars depicting +/- 2 s.e. The bottom left subfigure does the same but for the height of the infestation signs. Thus, for example, the distribution of emergency holes on NW orientation in Figure 11 has a mean of approximately 0.3 holes per tree with a confidence interval ≈0.06-0.54. The subfigures on the right side summarize results for stochastic dominance in the distribution of the given damage. All columns of the matrices represent the results of the pairwise Mann-Whitney test. The dark shading indicates that the distribution of damage with an orientation or height specified in the respective row dominates the distribution in the corresponding column with a specified significance level. Asterisks ***, **, and * denote 1%, 5%, and 10% significance, respectively. Returning to the example in Figure 11, the number of emergency holes with the SW orientation is statistically higher than the one with the NW orientation, and this result is significant at the 5% significance level.”

We also improved the resolution of the four figures in the revised manuscript.

Comment 8: It is standard to use tree diameter at breast height (dbh) in 5 or 10 cm dbh classes rather than perimeter.

Indeed, DBH (Diameter at breast height) is one of the most common dendrometric measurements. Yet, the height of an adult's breast is defined differently in different countries and situations. We preferred to do it at 1.3 m above ground, which is the one used in many countries. Also, since we did not have a tree caliper to measure the diameter at breast height of a tree, we used a girthing tape that measured the girth (circumference or perimeter) of the tree at breast height. While doing so we made sure the tape was perfectly level and that the tape was not kinked, so as not to skew the readings. We could have later calculated the diameters from the circumference readings, but decided to keep using the initial readings, i.e the circumference or perimeter of trees. Obviously, this would not affect the results. In Spain, companies applying endotherapy to inject trees also use the trunk perimeter at a height of 130 cm to determine the number of drillings.

Comment 9: It would be helpful to include a figure or table showing average ± SE number of exit holes, bark injuries, gallery slits at the time (or soon after, i.e., April or June 2018) of abamectin treatment for treated and control trees and again in December 2018 along with percent reduction and statistical differences.

We believe that our analysis is misunderstood here. As we describe in the paper, the spread of the invasive species is a dynamic process and depends on many factors such as the number of infestation signs already observed, including those of the different type (e.g. number of new gallery slits depending on earlier number of bark injuries or emergence holes). Hence, it would be incorrect to simply compare the number of infestation signs before and after treatment expecting the numbers to reduce due to the treatment. Instead, what we do in the regression analysis is to compare the spread of infestation signs of the treated trees to those trees that did not receive any abamectin treatment (control group) to infer if the treatment had any significant effect. One would not come to the same conclusion by just looking on the number of infestation signs before and after the treatment.

What we tried to do is to provide the averages (+/-2 s.e.) for the treated and the control group in the table below to see if we observe interesting and significant results. However, as we have many observations with zero infestation signs in June 2018, we cannot calculate percentage changes, but only absolute changes. As you can see from the table below, absolute increases in the number of infestation signs in the control group is always higher than in the treatment group even though in June 2018 the number of infestation signs of all three types was higher in the latter group. Since we do not control for the different starting conditions, the results of the Mann-Whitney test on statistical difference between absolute changes in the treated and the control group here are negative, i.e. no significant difference. But as we have shown in the regression analysis in the manuscript, controlling for the initial conditions and other factors like perimeter or height of the tree show that the abamectin treatment is significant. For the reasons abovementioned we prefer to not include the results from the table below in the manuscript.

 Sample of trees treated with abamectin (N=99) Control group created with propensity score matching (N=99) Mann-Whitney’s test p-value

 June

2018 December 2018 Absolute change June 2018 December 2018 Absolute change 

Emergence holes 2.04

[0.87-3.22] 2.86

[1.36-4.36] 0.82

 [0.37-1.27] 0.96

[0.28-1.64] 1.84

[0.92-2.76] 0.88

[0.36-1.40] 1

Bark injuries 0.54

[0.27-0.80] 0.62

[0.33-0.91] 0.08

[0.02-0.14] 0.39

[-0.05-0.83] 0.72

[0.24-1.20] 0.32

[0.11-0.54] 0.12

Gallery 

slits 3.61

[2.85-4.36] 3.72

[2.96-4.48] 0.11

[0.03-0.19] 2.17

[1.37-2.98] 2.31

[1.50-3.13] 0.14

[0.02-0.26] 0.64

Note: Results are reported in averages +/- 2 s.e.

Comment 10: It would also be helpful to include an overall summary/conclusion. How does the within tree distribution and abamectin treatment evaluation improve recommendations for survey and management?

We have included a short Conclusion section at the end of the manuscript, just before the Acknowledgments, as follows:

“We have evaluated how the infestation progressed as a function of past infestation and provided new relevant insights into female beetle’s preferences to lay their eggs on specific trees or specific areas of trees, therefore determining the within-tree distribution of larvae. The latter was done considering tree size, the geographical orientation of infestation damages and their location on trees, therefore providing key information to surveyors and managers on how to deal with this pest. Specifically, our statistical analysis shows that females prefer laying eggs on larger trees, on the highest part of trunks and on the crown base (this being more preferred than the trunk), and they do so on SW orientation avoiding those facing N, E and NW. We have also shown that emergence holes and gallery slits predict new coming infestation on trees. An abamectin treatment (trunk injection) carried out at the end of April significantly reduced the number of new infestation. However, managers should consider that, most likely, for maximum insecticide efficiency, the best time for treating with abamectin would be from mid-July to mid-August, when newly hatched larvae start feeding on the tree phloem.”

---

## [Editor Report · Decision Letter 1]

8 Dec 2020

PONE-D-20-28834R1

The invasive longhorn beetle Xylotrechus chinensis, pest of mulberries, in Europe: study on its local spread and efficacy of abamectin control

PLOS ONE

Dear Dr. SARTO I MONTEYS,

Thank you for submitting your manuscript to PLOS ONE. After careful consideration, we feel that it has merit but does not fully meet PLOS ONE’s publication criteria as it currently stands. Therefore, we invite you to submit a revised version of the manuscript that addresses the points raised during the review process.

As the authors have already addressed a thorough review of their manuscript, the editor has examined the revised version and suggests the changes listed below, which can be altered if the editor misunderstood the meaning of the authors:

Line 17 “ a potentially lethal pest of mulberry trees (Moraceae: Morus sp.)

Line 19 “the infestation spread”

Line 22 “and avenues is a concern, as beetle infestation “

Line 23 “how the infestation progresses over time, with and without Abamectin treatment, and provide insights into female egg-laying preferences.  Such knowledge helps contribute to management efforts to reduce expansion of the range of beetle infestation.”

Line 27 “do so on warmer, SW orientations rather than those facing N, NW and E. Emergence holes and gallery slits predict the spreading of infestations to new trees”

Line 30 “begin feeding on the phloem”

Line 45 Delete “As far as we know . . .described”

Line 53 “This invasive species is native”

Line 75 “into these French”

Line 87 delete “, as we report here”

Line96 “The data were assembled using ArcMap 10.3.1 . .”

Line 112 “Spain and other southern European countries.”

Line 114 Delete “Quite similar . . .countries.”

Line 120 “concern because beetle infestation increases the risk of falling branches and the need for rapid response by municipal authorities.”

Line 128 “observations, three types of indicators were selected to document infestation: (1) “

Line 130 “were counted and processed as described in the methods to inform the municipality about the progress of the beetle infestation in the study area.”

Line 156 “Occasionally, full-grown larvae have been found overwintering under”

Line 157 “Bark injuries become increasingly more evident “

Line 160 “Consequently, the presence of bark injuries can be a good”

Line 172 “which is bored into for pupation”

Line 178 “No studies were reviewed that reported how”

Line 180 “i.e., a statistical analysis of new infestation in relation to past infestationNor could we find studies about the . . . perimeter, or about the geographical”

Line 183 “provide important knowledge on the preferred trees for egg laying and where the eggs are most likely to be laid. A goal of the study was to determine such information to assist in scouting for infestations.

Line 230 “Specifically, to aid in scouting, it is important to know”

Line 234, or perhaps to wait until the discussion. Based on responses of other invasive beetle pests, one might also consider volatiles released by the trees, which, if were attractive would attract beetles over wide areas. The authors don’t bring this up, but many entomologists working in this research area would gravitate towards that possibility immediately. Potential attractants are not necessarily tree-based, so attractants of related longhorned beetles might be of interest.

Line 240 “In 2018, two surveys were conducted under permissions granted by the City Council, checking all public trees for beetle infestation.”

Line 262 “bark injuries (BI), noting also their heights above the ground and their compass orientations as one of 8 directions: N, NE, E, SE, S, SW,W, or NW.

Line 267, delete “To set . . .”

Line 287 “Because these 107 trees, among the 438 checked in June 2018, were still present in December, an analysis of the treatment results was included in the study.

Line 294 “were 2-3 drillings per tree.”

Line 311 “crown base, a non-parametric paired, 2-sample Wilcoxon Test was used that explicitly”

Line 315 delete “here”

Line 316 “A P-value below 0.05 leads to rejection of the null hypothesis of no difference between crown base and trunk damage.”

Line 319 “A statistical analysis of new infestations in December 2018 was conducted in relation to past”

Line 320 “Because the dependent variables are count data with many zeros”

Line 332 “A tree was considered to be infested when”

Line 415” Regression analysis was used to test if female beetles prefer to lay eggs in trees of specific heights. Tree height was not found to be a significant factor affecting the distribution of December damage compared to June (see below). Females of this beetle species are good flyers . . .”

Line 429 “Perimeter lengths were grouped into five ranges “

Line 443 “where the tree perimeter length had a “

Line 458 “This result does not have an obvious explanation”

Line 464 “Comparisons also were made of all damages”

Line 479 “As mentioned above, all 438 trees were in this analysis, including the December 2018 data and the data contributed by the 30 trees in June that had been removed by December.”

Line 484 “The goal was”

Line 490 and elsewhere, including line 496 “emergency” should be “emergence”

Line 506 The compass distribution of EH (Fig 11) indicates that females prefer to lay eggs on the SW side of the tree, both on the trunk and crown base. The 150-200-cm band and the 200-250-cm band” have higher values than the others. A somewhat similar pattern emerges in the distribution of BI”

Line 511 and elsewhere “popular” or “unpopular” should be replaced by “less preferred” or by a statement based on preference rather than popularity. Unfortunately, there are many words like this in the manuscript that should be replaced by more scientifically descriptive terminology or it will reflect negatively on the authors and the editor.

Line 525 Do not start a new paragraph here.

Line 526 new paragraph at “However, comparing the distribution . . “

Line 529 “This suggests that females prefer”

Line 532 and elsewhere, “Wilcox” should be “Wilcoxon”

Line 535 delete “Econometric”

Line 537 “The differences between December and June in the number of new tree damages. . .  was compared for height and perimeter, as well as the “

Line 539 “In addition the results were tested for the sum of BI +GS new damage.”

Line 572 “The results indicate that the number of GS in June is a significant predictor of “

Line 587 “This may be interpreted “. A statement like this is definitely discussion rather than results. While the editor considers this a plausible interpretation, there is not information about effects of chemical or visual stimuli to confirm the interpretation. It seems best to separate Results and Discussion, and then put statements like this only in the discussion. There are several other places also where it would benefit the manuscript to separate results from discussion.

Line 595 “Therefore, abamectin treatment was included as a predictor in models on the full sample and a robustness analysis was included, in which a propensity score matched treated trees against nearby untreated trees with respect to the five variables listed in Table 4.” ? The original sentence was not so clear.

Line 598 “As a result . . .” This sentence probably now should be deleted.

Line 601 Delete “As we see,”  Then say: “After matching of propensity scores, “

Line 602 “become close to” should be “approach”

Line 603 “further illustrate this, not just for the means, but for the entire”

Line 619 “the effects of abamectin were statistically significant, reducing the numbers of new infestations”

Line 627 “The larval distributions were mapped with respect to tree height  . . .”

We look forward to receiving your revised manuscript.

Kind regards,

Richard Mankin, Ph. D.

Academic Editor

PLOS ONE

---

## [Author Response · Author response to Decision Letter 1]

24 Dec 2020

Responses to the Editor

Title: "The invasive longhorn beetle Xylotrechus chinensis, pest of mulberries, in Europe: study on its local spread and efficacy of abamectin control”

Manuscript ID: PONE-D-20-28834 PLOS ONE

We do appreciate and are very grateful to the Editor for his good editing and valuable suggestions. Below we explain (in blue letters) how we have addressed such comments/suggestions.

Line 17 “a potentially lethal pest of mulberry trees (Moraceae: Morus sp.) Done!

Line 19 “the infestation spread” Done!

Line 22 “and avenues is a concern, as beetle infestation “ Done!

Line 23 “how the infestation progresses over time, with and without Abamectin treatment, and provide insights into female egg-laying preferences. Such knowledge helps contribute to management efforts to reduce expansion of the range of beetle infestation.” Done!

Line 27 “do so on warmer, SW orientations rather than those facing N, NW and E. Emergence holes and gallery slits predict the spreading of infestations to new trees” Done!

Line 30 “begin feeding on the phloem” Done!

Line 45 Delete “As far as we know . . .described” Done!

Line 53 “This invasive species is native” Done!

Line 75 “into these French” Done!

Line 87 delete “, as we report here” Done!

Line96 “The data were assembled using ArcMap 10.3.1 . .” Done!

Line 112 “Spain and other southern European countries.” Done!

Line 114 Delete “Quite similar . . .countries.” Done!

Line 120 “concern because beetle infestation increases the risk of falling branches and the need for rapid response by municipal authorities.” Done!

Line 128 “observations, three types of indicators were selected to document infestation: (1) “ Done!

Line 130 “were counted and processed as described in the methods to inform the municipality about the progress of the beetle infestation in the study area.” Done!

Line 156 “Occasionally, full-grown larvae have been found overwintering under” Done!

Line 157 “Bark injuries become increasingly more evident “Done!

Line 160 “Consequently, the presence of bark injuries can be a good” Done!

Line 172 “which is bored into for pupation” Done!

Line 178 “No studies were reviewed that reported how” Done!

Line 180 “i.e., a statistical analysis of new infestation in relation to past infestation Nor could we find studies about the . . . perimeter, or about the geographical” Done!

Line 183 “provide important knowledge on the preferred trees for egg laying and where the eggs are most likely to be laid. A goal of the study was to determine such information to assist in scouting for infestations. Done!

Line 230 “Specifically, to aid in scouting, it is important to know” Done!

Line 234, or perhaps to wait until the discussion. Based on responses of other invasive beetle pests, one might also consider volatiles released by the trees, which, if were attractive would attract beetles over wide areas. The authors don’t bring this up, but many entomologists working in this research area would gravitate towards that possibility immediately. Potential attractants are not necessarily tree-based, so attractants of related longhorned beetles might be of interest. 

Authors’ response: (Please, see also our response for Line 587 below). Indeed, what the Editor suggests is an important and interesting issue. Unfortunately, we did not work on that in our present work. However, we believe we should briefly mention these topics in our paper as something that should be studied in the future. Therefore, incorporating the Editor’s comments, we have added a brief paragraph at the end of the ‘Conclusions’ section that reads:

“Finally, there are issues which have not been brought up in this work but may be important for the management of this pest. Indeed, based on responses of other invasive beetle pests, research is needed to detect volatiles released by mulberries, because if they were attractive would attract beetles over wide areas. Likewise, since potential attractants are not necessarily tree-based, attractants released by related longhorned beetles might also be of interest.”

Because of this added paragraph, we have also changed the heading “Conclusions” for “Conclusions and future prospects”.

Line 240 “In 2018, two surveys were conducted under permissions granted by the City Council, checking all public trees for beetle infestation.” Done!

Line 262 “bark injuries (BI), noting also their heights above the ground and their compass orientations as one of 8 directions: N, NE, E, SE, S, SW, W, or NW. Done!

Line 267, delete “To set . . .” Done!

Line 287 “Because these 107 trees, among the 438 checked in June 2018, were still present in December, an analysis of the treatment results was included in the study. Done!

Line 294 “were 2-3 drillings per tree.” Done!

Line 311 “crown base, a non-parametric paired, 2-sample Wilcoxon Test was used that explicitly” Done!

Line 315 delete “here” Done!

Line 316 “A P-value below 0.05 leads to rejection of the null hypothesis of no difference between crown base and trunk damage.” Done!

Line 319 “A statistical analysis of new infestations in December 2018 was conducted in relation to past” Done!

Line 320 “Because the dependent variables are count data with many zeros” Done!

Line 332 “A tree was considered to be infested when” Done!

Line 415” Regression analysis was used to test if female beetles prefer to lay eggs in trees of specific heights. Tree height was not found to be a significant factor affecting the distribution of December damage compared to June (see below). Females of this beetle species are good flyers . . .” Done!

Line 429 “Perimeter lengths were grouped into five ranges “Done!

Line 443 “where the tree perimeter length had a “Done!

Line 458 “This result does not have an obvious explanation” Done!

Line 464 “Comparisons also were made of all damages” Done!

Line 479 “As mentioned above, all 438 trees were in this analysis, including the December 2018 data and the data contributed by the 30 trees in June that had been removed by December.”

Line 484 “The goal was” Done!

Line 490 and elsewhere, including line 496 “emergency” should be “emergence” Done!

Line 506 The compass distribution of EH (Fig 11) indicates that females prefer to lay eggs on the SW side of the tree, both on the trunk and crown base. The 150-200-cm band and the 200-250-cm band” have higher values than the others. A somewhat similar pattern emerges in the distribution of BI” Done!

Line 511 and elsewhere “popular” or “unpopular” should be replaced by “less preferred” or by a statement based on preference rather than popularity. Unfortunately, there are many words like this in the manuscript that should be replaced by more scientifically descriptive terminology or it will reflect negatively on the authors and the editor.

Authors’ response: We have addressed this point wherever necessary and believe now it reads much better.

Line 525 Do not start a new paragraph here. Done!

Line 526 new paragraph at “However, comparing the distribution... “Done!

Line 529 “This suggests that females prefer” Done!

Line 532 and elsewhere, “Wilcox” should be “Wilcoxon” Done!

Line 535 delete “Econometric” Done!

Line 537 “The differences between December and June in the number of new tree damages… was compared for height and perimeter, as well as the “

Authors’ response: The suggested sentence does not reflect exactly what we meant since here we did not compare differences between June and December but carried out a NBRM regression analysis. Therefore, we have modified and simplified the sentence as follows (which we believe it reflects exactly what we did): “By using a NBRM regression analysis, the number of new tree damages in December was explained by the damage the trees showed in June, their height and perimeter, as well as the abamectin treatment 107 trees received on April 24, 2018.”

Line 539 “In addition the results were tested for the sum of BI +GS new damage.” Done!

Line 572 “The results indicate that the number of GS in June is a significant predictor of “Done!

Line 587 “This may be interpreted “. A statement like this is definitely discussion rather than results. While the editor considers this a plausible interpretation, there is not information about effects of chemical or visual stimuli to confirm the interpretation. It seems best to separate Results and Discussion, and then put statements like this only in the discussion. There are several other places also where it would benefit the manuscript to separate results from discussion.

Authors’ response: We understand the Editor’s point. However, in this paper we believe it is better to use the option “Results and discussion” instead of the option “Results” and then “Discussion”. This is because this is a rather complex study, where we are dealing with at least six groups of results (*Infestation of public mulberry trees from 2016 to 2018; *Infestation with respect to trunk height; *Infestation with respect to trunk perimeter; *Orientation and location of infestation damages; *Analysis of new infestation in December 2018 as a function of past infestation and other tree variables; *Effects of abamectin treatment), and to our understanding confining specific discussion (if needed) within each specific topic adds clarity to the paper.

Concerning our statement that “This may be interpreted as beetles reducing infestation of a severely infested tree and possibly moving to a different one.“, it is true that we have no information about effects of chemical or visual stimuli to confirm the interpretation. Although this topic is undoubtedly very interesting and we might look into it in the future, for the time being we could not say much more than what is said in this short statement. But we have changed it as follows: “This may be interpreted as beetles reducing infestation of a severely infested tree and possibly moving to a different one, although to confirm it new research is needed about the effects of chemical and visual stimuli.”

Line 595 “Therefore, abamectin treatment was included as a predictor in models on the full sample and a robustness analysis was included, in which a propensity score matched treated trees against nearby untreated trees with respect to the five variables listed in Table 4.” ? The original sentence was not so clear. Done!

Line 598 “As a result . . .” This sentence probably now should be deleted.

Authors’ response: No, we should keep the sentence, because here we explain that we obtained a subsample of 198 trees. Thus, it contains new information. Keep in mind that a total of 107 trees were treated with abamectin, from which only 99 were selected.

Line 601 Delete “As we see,” Then say: “After matching of propensity scores, “Done!

Line 602 “become close to” should be “approach” Done!

Line 603 “further illustrate this, not just for the means, but for the entire” Done!

Line 619 “the effects of abamectin were statistically significant, reducing the numbers of new infestations” Done!

Line 627 “The larval distributions were mapped with respect to tree height . . .”

Authors’ response: We have changed and simplified the original paragraph as follows:

“We have evaluated how the infestation progressed as a function of past infestation and provided new relevant insights into female beetle’s preferences to lay their eggs on specific trees or specific areas of trees, hence mapping the larval distributions in them. This mapping was done by locating the infestation damages observed on the trees (their height and compass orientation), therefore providing key information to surveyors and managers on how to deal with this pest.”

---

## [Editor Report · Decision Letter 2]

4 Jan 2021

The invasive longhorn beetle Xylotrechus chinensis, pest of mulberries, in Europe: study on its local spread and efficacy of abamectin control

PONE-D-20-28834R2

Dear Dr. SARTO I MONTEYS,

We’re pleased to inform you that your manuscript has been judged scientifically suitable for publication and will be formally accepted for publication once it meets all outstanding technical requirements.

Kind regards,

Richard Mankin, Ph. D.

Academic Editor

PLOS ONE
---

## [Editor Report · Acceptance letter]

8 Jan 2021

PONE-D-20-28834R2 

The invasive longhorn beetle *Xylotrechus chinensis*, pest of mulberries, in Europe: study on its local spread and efficacy of abamectin control 

Dear Dr. Sarto i Monteys:

I'm pleased to inform you that your manuscript has been deemed suitable for publication in PLOS ONE. Congratulations! Your manuscript is now with our production department. 

Kind regards, 

on behalf of

Dr. Richard Mankin 

Academic Editor

PLOS ONE